# Chromatin profiling reveals heterogeneity in clinical isolates of the human pathogen *Aspergillus fumigatus*

Ana Cristina Colabardini[1,2☯], Fang Wang[2,3☯]*, Zhengqiang Miao[2☯], Lakhansing Pardeshi[2,4], Clara Valero[1], Patrícia Alves de Castro[1], Daniel Yuri Akiyama[5], Kaeling Tan[2,4], Luisa Czamanski Nora[6], Rafael Silva-Rocha[6], Marina Marcet-Houben[7,8], Toni Gabaldón[7,8,9], Taicia Fill[5], Koon Ho Wong[2,10,11]*, Gustavo H. Goldman[1]*

1 Faculdade de Ciências Farmacêuticas de Ribeirão Preto, Universidade de São Paulo, Ribeirão Preto, São Paulo, Brazil, 2 Faculty of Health Sciences, University of Macau, Macau SAR of China, 3 Intensive Care Unit, Biomedical Research Center, Shenzhen Institute of Translational Medicine, Shenzhen Second People's Hospital, The First Affiliated Hospital of Shenzhen University, Shenzhen, China, 4 Genomics, Bioinformatics and Single Cell Analysis Core, Faculty of Health Sciences, University of Macau, Macau SAR of China, 5 Instituto de Química, Universidade Estadual de Campinas, Campinas, São Paulo, Brazil, 6 Faculdade de Medicina de Ribeirão Preto, Universidade de São Paulo, Ribeirão Preto, São Paulo, Brazil, 7 Barcelona Supercomputing Centre (BSC-CNS). Jordi Girona, Barcelona, Spain, 8 Institute for Research in Biomedicine (IRB Barcelona), The Barcelona Institute of Science and Technology, Baldiri Reixac, Barcelona, Spain, 9 Catalan Institution for Research and Advanced Studies (ICREA), Barcelona, Spain, 10 Institute of Translational Medicine, Faculty of Health Sciences, University of Macau, Macau SAR of China, 11 MoE Frontiers Science Center for Precision Oncology, University of Macau, Macau SAR of China

☯ These authors contributed equally to this work.
* yb57660@connect.um.edu.mo (FW); koonhowong@um.edu.mo (KHW); ggoldman@usp.br (GHG)

**Data Availability Statement:** DNA sequencing and ChIP-seq data are available from the NCBI SRA database (accession number PRJNA728823, PRJNA728851 and PRJNA793345).

## Abstract

Invasive Pulmonary Aspergillosis, which is caused by the filamentous fungus *Aspergillus fumigatus*, is a life-threatening infection for immunosuppressed patients. Chromatin structure regulation is important for genome stability maintenance and has the potential to drive genome rearrangements and affect virulence and pathogenesis of pathogens. Here, we performed the first *A. fumigatus* global chromatin profiling of two histone modifications, H3K4me3 and H3K9me3, focusing on the two most investigated *A. fumigatus* clinical isolates, Af293 and CEA17. In eukaryotes, H3K4me3 is associated with active transcription, while H3K9me3 often marks silent genes, DNA repeats, and transposons. We found that H3K4me3 deposition is similar between the two isolates, while H3K9me3 is more variable and does not always represent transcriptional silencing. Our work uncovered striking differences in the number, locations, and expression of transposable elements between Af293 and CEA17, and the differences are correlated with H3K9me3 modifications and higher genomic variations among strains of Af293 background. Moreover, we further showed that the Af293 strains from different laboratories actually differ in their genome contents and found a frequently lost region in chromosome VIII. For one such Af293 variant, we identified the chromosomal changes and demonstrated their impacts on its secondary metabolites production, growth and virulence. Overall, our findings not only emphasize the influence of

**Funding:** The Science and Technology Development Fund, Macao S.A.R (FDCT) (project reference number: 0106/2020/A and 0033/2021/A1) to KHW, Fundação de Amparo à Pesquisa do Estado de São Paulo (FAPESP) 2016/07870-9 to GHG and 2018/14821-0 to ACC and Conselho Nacional de Desenvolvimento Científico e Tecnológico (CNPq) (301058/2019-9 and 404735/2018-5) to GHG. The funders had no role in study design, data collection and analysis, decision to publish, or preparation of the manuscript.

**Competing interests:** The authors have declared that no competing interests exist.

genome heterogeneity on *A. fumigatus* fitness, but also caution about unnoticed chromosomal variations among common laboratory strains.

---

## Author summary

The fungal pathogen *A. fumigatus* has a cosmopolitan distribution, which make its isolates genotypically and phenotypically diverse. The two commonly used reference clinical isolates, Af293 and CEA17, differ in growth, virulence and susceptibility to the immune system and drug treatment; however, their genomes are 99.8% identical with differences mainly at non-coding regions. The high genetic similarity between the two isolates implicates epigenetic controls for their physiological differences. Here, we analysed two chromatin modifications in the *A. fumigatus* genome and found that H3K4me3 profiles of the two isolates are similar, while H3K9me3 modification at the genome-wide level is highly variable. The H3K9me3 differences between Af293 and CEA17 are correlated with differences in transposon silencing and inactivation. In support of this, we observed independent gross chromosomal alterations in the Af293 strains of different laboratories, but not among CEA17 strains. Characterization of one of the Af293 variant strains, which has undergone chromosomal loss and amplification of two separate regions in chromosome VIII (Chr VIII), showed significant improvements in secondary metabolites production, growth and virulence compared to the original Af293 isolate. These results suggest that stochastic gross chromosomal changes occur more frequently in certain isolates. Overall, this study provides a link between epigenetic and genetic mechanisms for the adaptation of *A. fumigatus* to diverse environments.

## Introduction

Invasive Pulmonary Aspergillosis (IPA) is a life-threatening infection in immunosuppressed patients [1,2] caused mainly by the filamentous fungus *Aspergillus fumigatus*, a ubiquitous soil inhabitant that can utilize a wide variety of organic substrates and grow under a broad range of environmental conditions [3]. *A. fumigatus* produces asexual conidia that readily become airborne and can survive diverse environmental conditions [4]. Several *A. fumigatus* strains can infect immunosuppressed patients [2]. Once inhaled by a human host, *A. fumigatus* conidia penetrate deep in the alveoli, where they must survive the surveillance of the host immune system [1,2,5]. Alveolar macrophages kill conidia and germlings within the phagolysosome by producing reactive oxygen species (ROS) and phagolysosomal acidification [6], while neutrophils can attach to germlings and release granules containing a variety of antimicrobial compounds [6,7]. If the invader succeeds in establishing an infection in a host who is then subjected to an antifungal treatment, *A. fumigatus* hyphae must also survive the fungicidal and/or fungistatic effects of the antifungal drug(s) to persist inside the host [8]. Thus, it is essential that *A. fumigatus* can adapt its physiology to the changing conditions during the course of infection to thrive as a pathogen [9].

Genomic plasticity is a key factor for the success of pathogenic fungi, as it can rewire transcriptional events [10] and/or generate variability that allow organisms to adapt to different environments [11] and to improve their arsenal of defences against the host [9,12]. Fungal variants with genotypic diversity can be generated through mitotic [13,14] and meiotic [15] processes during development in nature, resulting in isolates with genetic differences ranging

from single-nucleotide polymorphisms to chromosome rearrangements [16,17]. It has been shown in the fungal pathogen *Candida albicans* that chromosome rearrangements are driven by long repeat sequences [18,19] and subjected to epigenetic control [20].

Epigenetic regulation is coordinated by the chromatin, which is composed of DNA wrapped around nucleosomes and is essential for genome compaction and maintenance [21]. The tail of histone proteins in nucleosomes are subjected to various post-translational modifications (PTMs), which plays influential roles towards the control of chromatin status and activities [22]. Euchromatin (which is generally composed of gene-rich and non-repetitive DNAs) and heterochromatin (often consists of regions with low gene densities and repetitive DNAs) are usually marked by different modifications, such as trimethylation of lysines 4 and 9 on histone H3, respectively [23,24]. At the chromosomal level, the centromeric and telomeric regions (which contains characteristic DNA repeats) have important roles in kinetochore function [25] and protection of chromosome ends [26], respectively, while subtelomeres contain silent genes intermingled with conditional active genes, such as those for secondary metabolites (SM) production [27–29]. Several histone PTMs were described in filamentous fungal species, such as *A. nidulans* [30,31], *A. oryzae* [32,33], *A. flavus* [34], *Neurospora crassa* [35], *Fusarium fujikuroi* [36,37] and *F. graminearum* [38] with distinct modification patterns and functions. In *A. fumigatus*, chromatin modifications have been shown to affect growth, conidiation, stress responses, biofilm formation and fitness during infection [39–41]. However, comprehensive analysis of specific PTMs of histones has never been carried out in this organism.

Transposable elements (TEs) play a significant role in genome evolution through the generation of variability and genetic instability [42]. TEs can multiply and/or transpose in the genome, and as a result, generate a myriad of effects including gene duplication and inactivation, disruption of regulatory regions, generation of alternative splicing variants and spreading of epigenetic silencing [43]. Furthermore, TEs, which contain repeated sequences, can passively act as substrate for ectopic recombination [42]. There are two main classes of TEs: i) Class I TEs are retrotransposons that use an intermediary RNA molecule and a reverse transcriptase to transpose, while ii) class II TEs are DNA transposons that encode transposases to directly mobilize DNA repeats [44]. TEs are normally silenced by heterochromatin [24] and may be activated or derepressed under stress [42]. Fungi often lose or acquire SM gene clusters to alter their arsenals for fitness and survival under specific environmental niches [45]. It has been shown that many acquired SM gene clusters in *Aspergillus* species are flanked by TEs, implying that TEs activity contributed to the acquisition of these important virulence factors [46]. Besides TEs, genetic and genomic changes can also originate from chromosome segmentation [9], which can facilitate strain adaptation and has been extensively studied in *C. albicans* [47–51]. The *A. fumigatus* genome contains an abundance of TEs [12,52] and high heterogeneity across different *A. fumigatus* isolates in drug susceptibility, nutrients acquisition and virulence has been reported [53–58]. Hence, it is interesting to understand whether and how those elements affect the physiology and adaptation of this pathogen to diverse environmental pressures during its evolution.

Af293 and CEA17 (a pyrimidine auxotrophic derivative of CEA10 [59]) are the two most common *A. fumigatus* clinical isolates used in the laboratory as references [52,60]. However, they bear significant differences in their physiology and virulence [61]. Recently, many factors, such as carbon and nitrogen metabolism and response to low oxygen stress in the lung, have been shown to be heterogeneous across these and other clinical isolates with impacts on their host colonization abilities [54–58]. Genomic comparisons have shown that Af293 and CEA17 have 98% overall genomic identity with the most variable regions located within 300 kb from the chromosomal telomeres [52]. The Af293 and CEA17 unique regions each include around

200 genes, and the majority of unique genes are clustered together in blocks ranging from 10 to 400 kb [52,62]. To address whether epigenetic factors also contribute to the variability of *A. fumigatus* isolates, we compared the well-established active (H3K4me3) and repressive (H3K9me3) histone codes in Af293 and CEA17. This work shows that the modification pattern of the histone marker H3K4me3 associated with active gene expression is similar between the two isolates, while the genomic regions marked by the repressive heterochromatic marker H3K9me3 are variable. Furthermore, we detected key differences in TEs distribution between the two isolates, with Af293 having a significantly higher number of RNA TE from the LINE family. We also uncovered that Af293 strains from different laboratories are variable in genome contents and identified a chromosomal loss hotspot. Characterization of an Af293 variant (which lost the hotspot region and had an amplification of another region at the left arm of the same chromosome) revealed increased fitness, mycotoxin production and virulence. Thus, our chromatin profiling analysis have revealed genetic and epigenetic heterogeneities between *A. fumigatus* isolates and demonstrated their influences on fitness.

## Results

### Active *A. fumigatus* genes undergo nucleosome depletion at promoters and H3K4me3 modification at 5' coding regions

In order to compare chromatin activities of the two *A. fumigatus* reference strains Af293 and CEA17, we performed Chromatin Immunoprecipitation coupled to high throughput sequencing (ChIP-seq) to map genome wide nucleosome occupancy (histone H3) and histone H3 K4 trimethylation (H3K4me3) that associates with active transcription. Input and immunoprecipitated DNA samples were sequenced using the Illumina HiSeq2500 platform and the sequencing reads were aligned to the reference genomes of Af293 or CEA17. The CEA17 reference genome has 55 contigs [52], while there are eight full chromosomal assemblies in the Af293 reference genome [60,63]. Comparison of the size of CEA17 reference genome (A1163) with those of Af293 and other *A. fumigatus* isolates (n = 79 from [64] and [65]) shows the CEA17 reference genome size is greater than most isolates (S1A Fig), and comparison to the full assembly of Af293 genome suggests that the CEA17 reference probably covers more than 99% of *A. fumigatus* genome (S1 Table), and therefore, has sufficient coverage for analysis and comparison. Nevertheless, since the Af293 reference has relatively better gene model and function annotations, we explored the feasibility of using the Af293 reference genome for CEA17 data analysis. Considering the fact that there are some minor differences between the two genomes [52], we mapped the CEA17 data to both CEA17 and Af293 reference genomes for a comparison and found only a marginal difference in terms of the mapping percentage (97.9% and 95.1% of CEA17 reads mapping to the CEA17 and Af293 reference genome, respectively) (S2 Table). This suggests that for most parts the use of Af293 reference genome for CEA17 data would have a small effect, if any, on the results and data interpretation. In light of this, results based on mapping to the Af293 genome will be presented, except for analyses whereby genome structure matters (in these cases the CEA17 reference is used and duly indicated).

Consistent with observations in other eukaryotes, promoters (between -500 to +100 bp with respect to annotated transcription start sites [TSS]) of *A. fumigatus* protein coding genes were found depleted for nucleosomes (Fig 1A), while H3K4me3 modifications were enriched at the first half of the open reading frame (ORF) of many genes (Fig 1B). To understand the relationship between promoter nucleosome occupancy, H3K4me3 modification and transcriptional activity, we also performed RNA sequencing (RNA-seq) for the two isolates under the same growth conditions (Bioproject SRP154134, [66]), and correlated nucleosome and H3K4me3 levels with mRNA abundance. We divided Af293 annotated genes into 5 groups

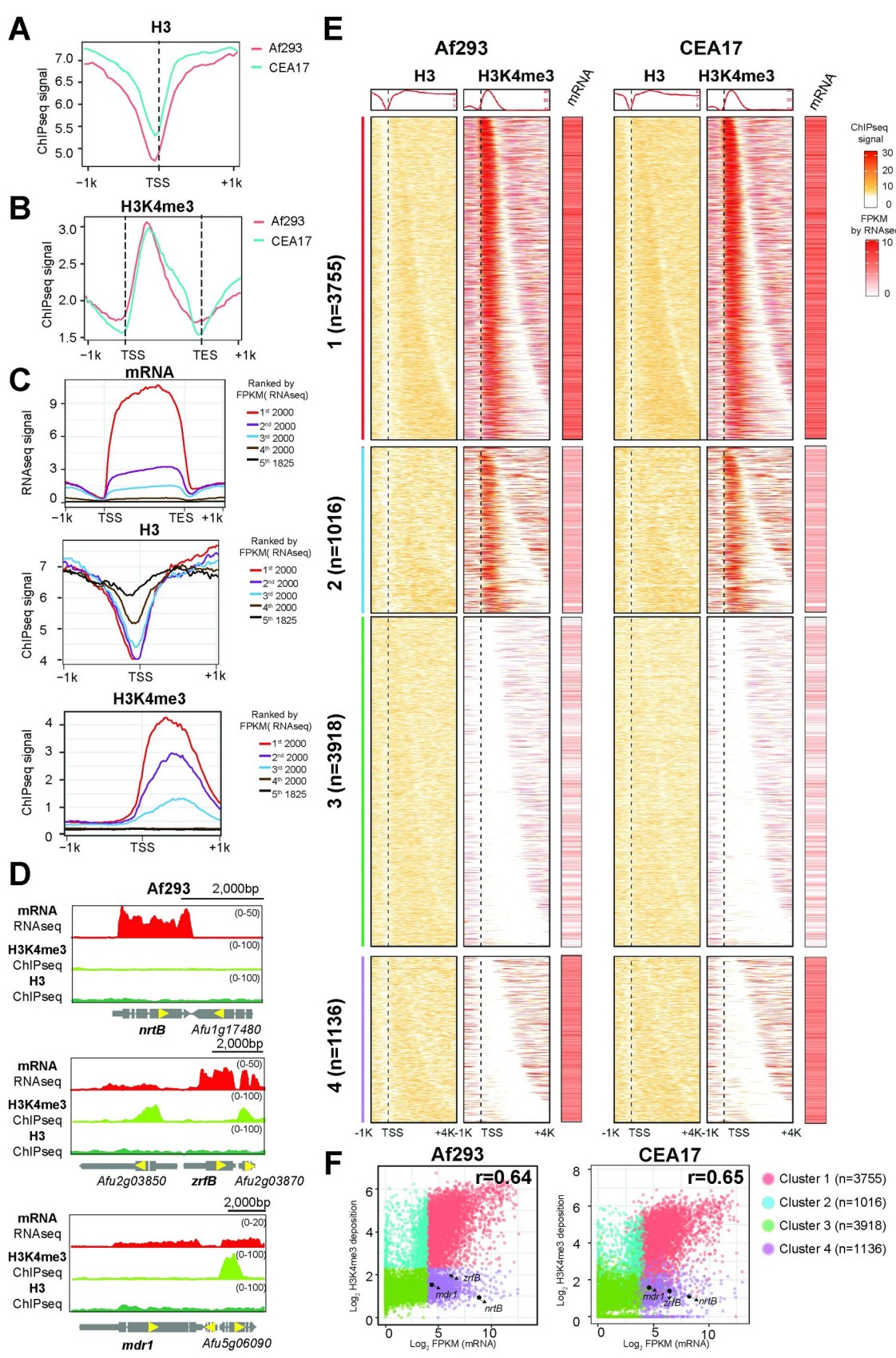

**Fig 1. The nucleosome depletion at promoters and H3K4me3 modification at 5' coding regions of two *A. fumigatus* isolates Af293 and CEA17.** (A) Line plots showing the H3 deposition within 1 kb of TSS region of genes genome wide. (B) Line plots showing the H3K4me3 deposition within 1 kb of CDS region of genes genome wide. (C) Line plots showing the mRNA level (top), and deposition of H3 (middle) and H3K4me3 (bottom) within 1 kb of CDS or TSS regions of genes genome wide. Gene order is ranked by mRNA level from high to low. (D) Genome browser screenshots showing mRNA and H3K4me3 level at selected genes. H3 level was used as control. (E) Heatmap plots showing the mRNA accumulation and deposition of H3 and H3K4me3 in Af293 and CEA17 genes genome wide classified as four clusters. (F) Scatter plots showing the mRNA and H3K4me3 levels of the four cluster genes as shown in (E).

based on their gene expression levels determined by RNA-seq from the most (1st 2000) to the least expressed (5th 1825) (Fig 1C, top panel). ChIP-seq analysis of histone H3 occupancy revealed that the two most expressed gene groups had the lowest H3 accumulation at the TSSs in Af293 (Fig 1C, middle panel), which suggests that nucleosomes were depleted at these promoters (*i.e.*, a permissive state for transcription). On the other hand, the expression levels of genes in the 3rd, 4th and 5th groups were inversely proportional to their promoter H3 levels (Fig 1C, middle panel), reflecting a general negative relationship between promoter nucleosome density and transcriptional activity, consistent with the well-established role of nucleosome occupancy and positioning in controlling transcriptional activation.

In contrast, H3K4me3 modification is positively correlated with transcriptional activity in which the most expressed genes had high levels of H3K4me3 modification at the 5' end of ORFs, and the levels reduced proportionally in the second and third groups (Fig 1C, bottom panel). The correlated gene expression and H3K4me3 suggests a positive association between H3K4me3 modification and transcriptional activity in Af293, agreeing with published literature of other eukaryotes [23]. For the CEA17 isolate, the same correlations of H3 and H3K4me3 with transcriptional activity (mRNA levels by RNA-seq) were observed, regardless of which reference genome was used for the analysis (S1B Fig). Global gene expression levels (*e.g.*, FPKM from RNA-seq) and H3K4me3 ChIP-seq profiles were also highly correlated between the Af293 and CEA17 isolates (S1C Fig). Taken together, these observations indicate negative and positive effects of promoter nucleosome and H3K4me3 modification on transcriptional activity, respectively, in *A. fumigatus* and these chromatin regulations are general in the two different clinical isolates.

## Many expressed genes are devoid of H3K4me3 modification

In the course of data inspection on a genome-browser, we noted many well-expressed genes (*e.g. nrtB*, *zrfB*, and *mdr1*) without H3K4me3 modification at their coding regions for both Af293 (Fig 1D) and CEA17 (S1D Fig), suggesting a potentially interesting difference in the mechanistic relationship between transcription and H3K4me3 modification. To systematically analyse this, four gene clusters were classified according to the level of H3K4me3 modification and mRNA expression levels (Fig 1E). Genes in cluster 1 (n = 3755) had high H3K4me3 modification and mRNA levels (determined by RNA-seq) (Fig 1E and 1F). On the other hand, genes in cluster 2 (n = 1016) had relatively lower mRNA levels when compared to cluster 1, but significant amount of H3K4me3 ChIP-seq signals could still be observed (Fig 1E and 1F). Cluster 3 genes (n = 3918) had low or background levels of H3K4me3 and mRNA, which presumably is indicative of limited transcriptional activities (Fig 1E and 1F). Interestingly, a large group of well-expressed genes (n = 1136; cluster 4) showed a lack of H3K4me3 modification, similar to what was observed at the genome-browser (Fig 1E and 1F). The same observation was found for CEA17 (Fig 1E and 1F), suggesting that this is a general phenomenon for *A. fumigatus*.

Gene Ontology (GO) enrichment analysis revealed that cluster 1 genes are related to primary metabolic processes such as translation, cellular component organization and biosynthesis (S1E Fig and S1 Data), while genes in cluster 2 were mainly associated to DNA metabolism, DNA repair, tRNA processing, autophagy and melanin biosynthesis (S1E Fig and S1 Data). As expected, the lowly or non-expressed genes in cluster 3 were highly enriched with the secondary metabolism processes (S1E Fig and S1 Data), although cell wall remodelling genes were also present in this cluster (S1 Data). Interestingly, those genes with an incongruous H3K4me3 modification and gene expression levels (cluster 4) encode proteins involved in transmembrane transport functions such as carbohydrate, ion, ammonium and drug transporters (S1E Fig and S1 Data), suggesting that genes involved in transport processes may have an atypical mechanistic relationship between transcription and H3K4me3 modification.

## H3K9me3 modifications are found at telomeric, subtelomeric and pericentromeric regions as well as other chromosomal locations

After evaluation of the active chromatin marker, we also compared the histone modification H3K9me3 using ChIP-seq to investigate the heterochromatin situation in the two isolates. At the genome browser level, high enrichments of H3K9me3 modification were found at (sub) telomeric regions as well as near annotated centromeres (*i.e.*, pericentromeric regions) (Fig 2A). When the CEA17 H3K9me3 ChIP-seq data was mapped to the Af293 reference genome, the modification pattern in CEA17 is similar to that of Af293 (S2A Fig) except for a few (sub) telomeric regions where the two isolates differ significantly as reported previously [52]. The differences at these (sub)telomeric regions are most likely due to poor mapping of CEA17 sequence reads to the Af293 reference genome. In order to confirm this and rule out any other potential mapping artefact due to genome differences, we then mapped the CEA17 data to its own reference genome. While a full analysis on all centromeric and telomeric regions is not possible for CEA17 data due to its partial assembled genome and poor annotation, H3K9me3 peaks were observed at both ends of the largest scaffolds (Scf_1, 2, 3 and 4) (S2B Fig), which are almost full chromosomal assemblies corresponding to Af293 chromosome I, II, III and V, respectively [52]. Therefore, H3K9me3 modified nucleosomes are deposited at (sub)telomeric regions in both Af293 and CEA17.

To systematically identify H3K9me3 modified regions throughout the genome, we applied MACS2 broad peaks calling method on the H3K9me3 ChIP-seq data of the two isolates mapped to their respective reference genome. A total of 121 and 143 H3K9me3 peaks were obtained for Af293 and CEA17, respectively (S2 Data). Most of them were located in non-coding regions (Fig 2B and S2 Data). Interestingly, for both isolates, the size of H3K9me3 modified regions (*i.e.*, ChIP-seq peaks) varied significantly, ranging from 0.2 to 45 kb (Fig 2C). Comparison between the two isolates showed that Af293 have more shorter H3K9me3 modified regions (*e.g.* <500 bp) than CEA17 (Fig 2D), indicating dissimilar H3K9me3 modification patterns in the two isolates.

We next mapped the 121 H3K9me3 peaks to the chromosomal assemblies of Af293 and found many are near the annotated centromeres and within 300 kb from chromosome ends (Fig 2E and S2 Data), which are referred as the sub-telomeric regions [67]. It is noteworthy that the majority of peaks (n = 82) were actually found distributed across chromosome bodies rather than at telomeric and subtelomeric regions (n = 39) (S2C Fig). Although the analysis for CEA17 was limited by its incomplete genome assembly, H3K9me3 peaks were clearly observed at telomeric and subtelomeric (n = 16) as well as internal regions (n = 45) of the four largest scaffolds (Scf_1, 2, 3 and 4). When comparing between the corresponding chromosomes of Af293 and CEA17, it was found that H3K9me3 peak locations are not the same at many

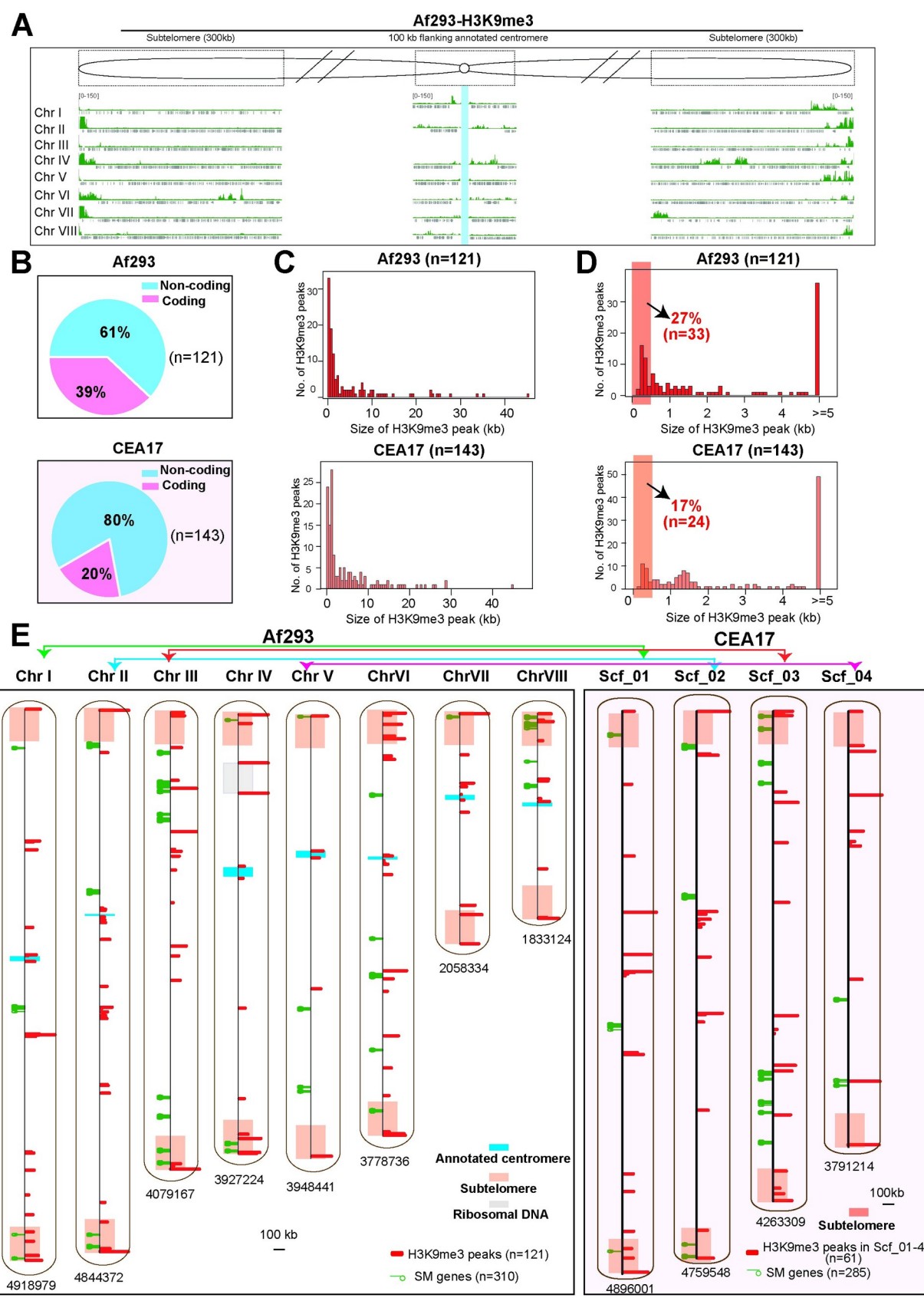

**Fig 2. Different H3K9me3 modification profiles in Af293 and CEA17.** (A) A overall genome browser screenshot showing the H3K9me3 ChIPseq signals in subtelomeric and pericentromeric regions in Af293 genome. Blue shade was marked with the annotated centromere region. (B) A sector diagram showing the genome wide distribution of H3K9me3 peaks called by MACS2 in Af293 and CEA17. (C-D) Histogram plots showing the size of H3K9me3 peaks in Af293 and CEA17 isolates. (E) A scheme representing the genome wide location of annotated centromeres (blue), subtelomeres (pink), rDNAs loci (grey), H3K9me3 binding peaks (red) and SM biosynthetic genes (green) along the 8 chromosomes of Af293 and 4 contigs of CEA17.

genomic regions (Figs 2E and S2D), suggesting dissimilar modifications on Af293 and CEA17 genomes.

## Most BGCs are not modified by the H3K9me3 heterochromatic modification in *A. fumigatus*

One of the major classes of genes suggested to be under H3K9me3 mediated transcriptional silencing is the secondary metabolite (SM) biosynthetic genes, which are often arranged in clusters (refer to as BGCs hereafter) [29]. Therefore, it is possible that the differences in H3K9me3 modifications in the two isolates are related to differences in SM BGCs regulation and/or genomic locations. As shown previously [52], the genomic locations of most SM BGCs on the four largest scaffolds of CEA17 (n = 17) were similar to the corresponding chromosomes I, II, III, V of Af293 (n = 19) (S3 Table), with a few exceptions (*e.g.*, BGCs 9–16 whose genomic arrangement was inverted between CEA17 Scf_3 and Af293 chromosome III (S3A Fig) and BGCs 27 and 33 that were located at different chromosomes in the two isolates (S3 Table). To better analyse the situation, we mapped all identified H3K9me3 peaks to SM BGCs in the two isolates. There are 31 common SM BGCs in both Af293 and CEA17, and Af293 genome contains two additional SM BGCs (BGC 1 and 4) that are not present in CEA17 (S3 Table) [46]. Surprisingly, we found that most BGCs are not associated with the detected H3K9me3 peaks (Fig 2E) with only two peaks located within SM BGCs in both Af293 (BGCs 10 and 16) (Fig 3A) and CEA17 (BGCs 16 and 21) (S3B Fig), even though the ChIP-seq experiment was carried out under the conditions in which most of the SM BGCs are silent. Moreover, the peaks found at SM BGCs only spanned a relatively small gene-free region (*e.g.*, about 1–8% of the size of BGC on average) (Figs 3A, S3B and S3C).

H3K9me3 has been associated with transcriptional silencing of the sterigmatocystin (ST) BGC genes in *A. nidulans* [68] and lolitrem and ergot alkaloid BGC genes in *Epichloë festucae* [69], and it was found that the modification occurs at the chromatic regions outside the gene clusters instead of over the promoter or body of BGC genes. We, therefore, extended the analysis to identify the nearest H3K9me3 peak for each SM BGC, and only 5 SM BGCs had an H3K9me3 peak located outside of and adjacent to the annotated cluster border in both Af293 (BGCs 4, 8, 19, 25 and 33) and CEA17 (BGCs 8, 10, 12, 19 and 25) (Fig 3B and 3C). Hence, together with the two BGCs with H3K9me3 peaks within the clusters (Figs 3A and S3B), only seven SM BGCs in each of the isolates (out of 33 and 31 SM BGCs in Af293 and CEA17, respectively) are modified by or located near H3K9me3 modified regions. Among these H3K9me3-associated clusters, five clusters (BGCs 8, 10, 16, 19 and 25) (Fig 3C) were common between the two isolates. Consistent with the heterochromatic role of H3K9me3, genes within these clusters have significantly lower expression (*i.e.*, silent) than those BGC genes without H3K9me3 modification in both isolates (Figs 3D and S3D). However, we also noted discordance between H3K9me3 modification and BGC silencing. For example, there were differential H3K9me3 modifications on BGCs 12, 21 and 33 between the two isolates (S3B Fig), but the BGCs were not silent (*i.e.*, mRNAs could be detected) and expression levels of their genes were highly similar between the two isolates (Fig 3E), suggesting that the observed H3K9me3 modifications did not have a strong transcriptional silencing effect, if any. In fact, overall expression

## A H3K9me3 modification within BGC

## B H3K9me3 modification outside BGC

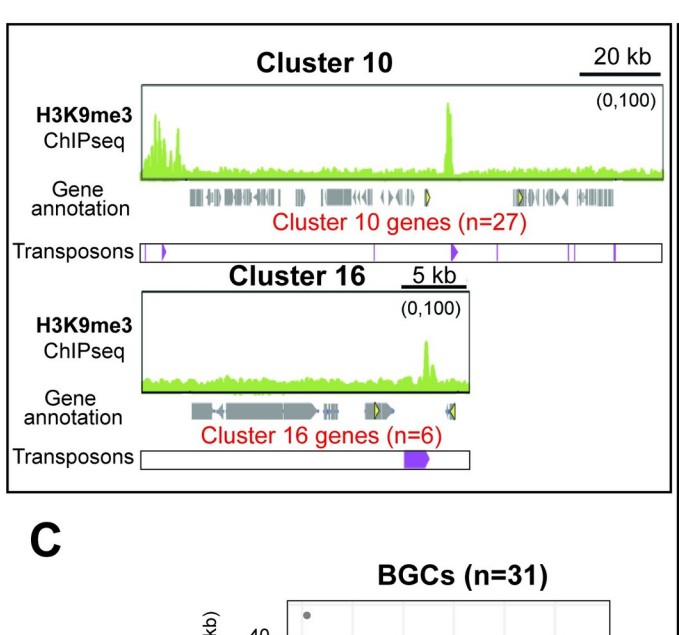

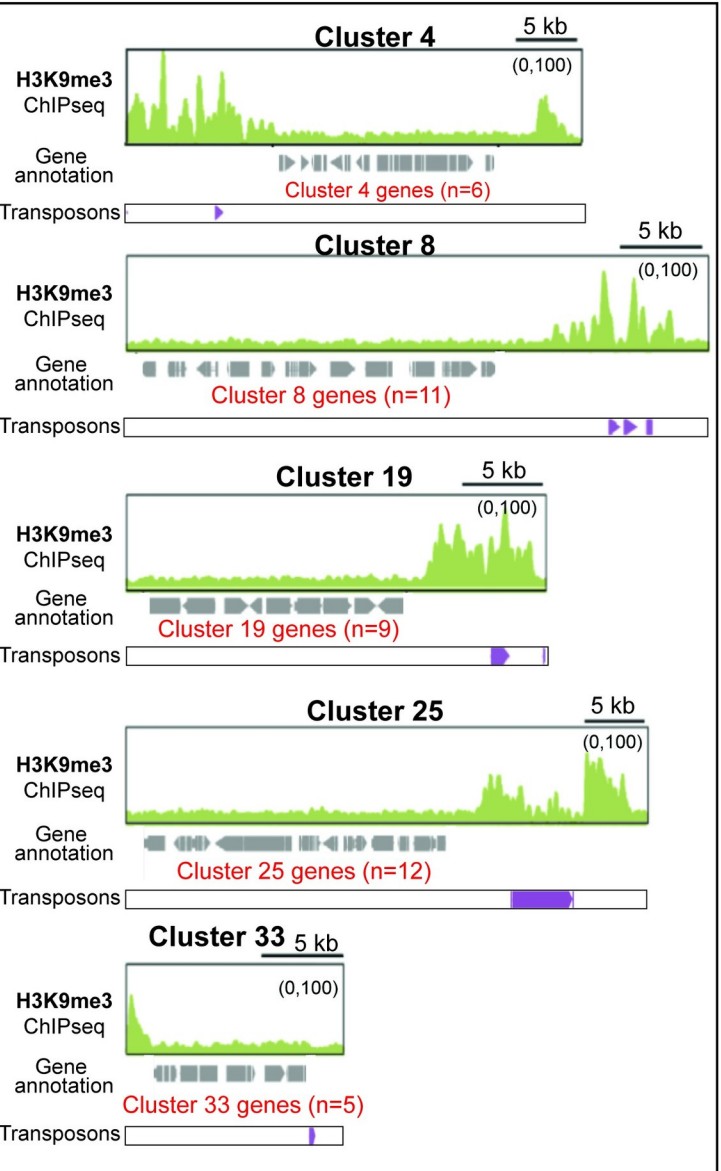

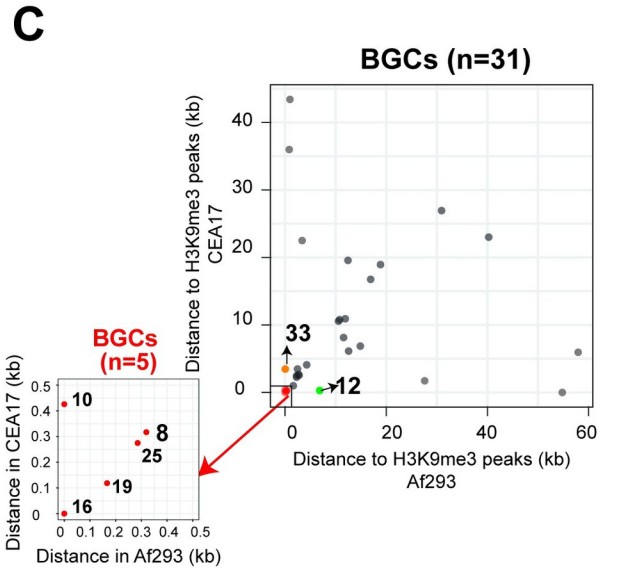

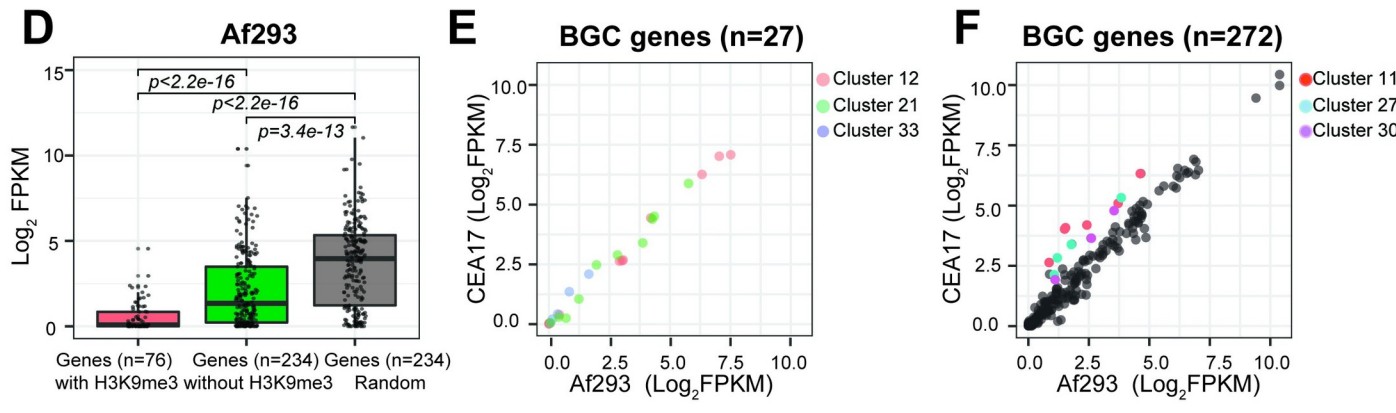

**Fig 3. Majority of BGCs in *A. fumigatus* were not marked by H3K9me3 in Af293 and CEA17.** (A) Genome browser screenshots showing the selected BGCs (BGC 10 and 16) marked with H3K9me3 in Af293. (B) Genome browser screenshots showing the selected BGCs marked with H3K9me3 within 5kb in Af293. TEs were indicated with purple markers in (A) and (B). (C) A scatter plot showing the distance of H3K9me3 peaks to BGCs in Af293 and CEA17. (D) A bar plot showing the expression of BGC genes with (red) or without (green) H3K9me3 in Af293. Random remaining non-SM genes (grey, n = 234) was plotted as control. (E) Scatter plots showing the expression of selected BGCs (12, 21, and 33) with distinct H3K9me3 modification in Af293 and CEA17. (F) Scatter plots showing the expression of selected BGCs with similar H3K9me3 modification in Af293 and CEA17. Genes in BGCs 11, 27 and 30 were coloured showing higher expression in CEA17 than Af293.

of BGC genes were highly correlated between the two isolates independent of whether their clusters were modified by H3K9me3 (Fig 3F), although they were generally lower than the average expression level of *A. fumigatus* genes (Figs 3D and S3D). Moreover, differential gene expression (Fig 3F) and SM production (S3E Fig) between the two isolates were noted for BGCs 11 (for fusarine C biosynthesis), 27 (for pyripyropene A biosynthesis) and 30 (for fumagillin biosynthesis); however, these clusters were not modified by or located near H3K9me3, suggesting that H3K9me3 modification may not be the major mechanism for silencing of SM genes. Taken together, we could not observe a clear link between H3K9me3 modification and BGCs silencing in *A. fumigatus*.

## Af293 and CEA17 genomes have distinct transposable element (TE) makeup

Since the BGCs are not the main reason for the observed H3K9me3 modification differences between the two isolates, we turned to TEs that are also targets of H3K9me3 heterochromatin regulation in many other organisms [24]. To systematically identify TEs, we employed the RepeatMasker program to search the Af293 and CEA17 reference genomes against an open database of TE profile HMM models and consensus sequences [70]. Since the CEA17 reference genome is not fully assembled, we first confirmed that the existing sequence and assembly is suitable for TE analysis. To this end, Illumina sequencing for the Af293 and CEA17 isolates was performed, and the raw sequences were mapped to both reference genomes separately. If there were many TEs missing from the partial CEA17 assembly, a high number of unmapped reads (from mapping CEA17 data to the CEA17 genome reference) is expected to contain TE sequences. When using the complete Af293 reference genome and Af293 data as a control, the comparison showed that the numbers of TEs from mapping the Af293 and CEA17 Illumina sequences to their respective genomes are similar (S4A Fig), indicating that the CEA17 reference genome is just as good as the Af293 one for TE analysis.

RepeatMasker analysis of the Af293 and CEA17 reference genomes identified 539 and 654 TEs, respectively (S4 Table). It is noteworthy that the number of TEs in both genomes far exceeded the number of H3K9me3 peaks (n = 121 and 143 for Af293 and CEA17, respectively), suggesting that not all TEs were modified by H3K9me3. Alternatively, each H3K9me3 modified region may overlap with multiple TEs. Indeed, some of the H3K9me3 peaks overlap with several TEs (Fig 4A and 4B) and strong H3K9me3 ChIP-seq signals were often found at or near TEs located in clusters (Fig 4B, indicated by arrow), while most isolated TEs have background levels of H3K9me3 (*i.e.*, no or non-detectable modification) (Fig 4C). Systematic analysis showed that about 40% of H3K9me3 peaks called from MACS2 were found at or near TE (s) (Fig 4D, left panel) and around half (45%) of TEs were associated with H3K9me3 modification (Fig 4D, right panel), when considering a window of 4 kb spanning the given feature. It is noteworthy that the H3K9me3 modifications found at some (but not all) BGCs also associated with the presence of TEs (Fig 3A and 3B), and different TE distribution may also explain the different H3K9me3 modification patterns at some BGCs between the two isolates (*e.g.* BGC21; S3B Fig).

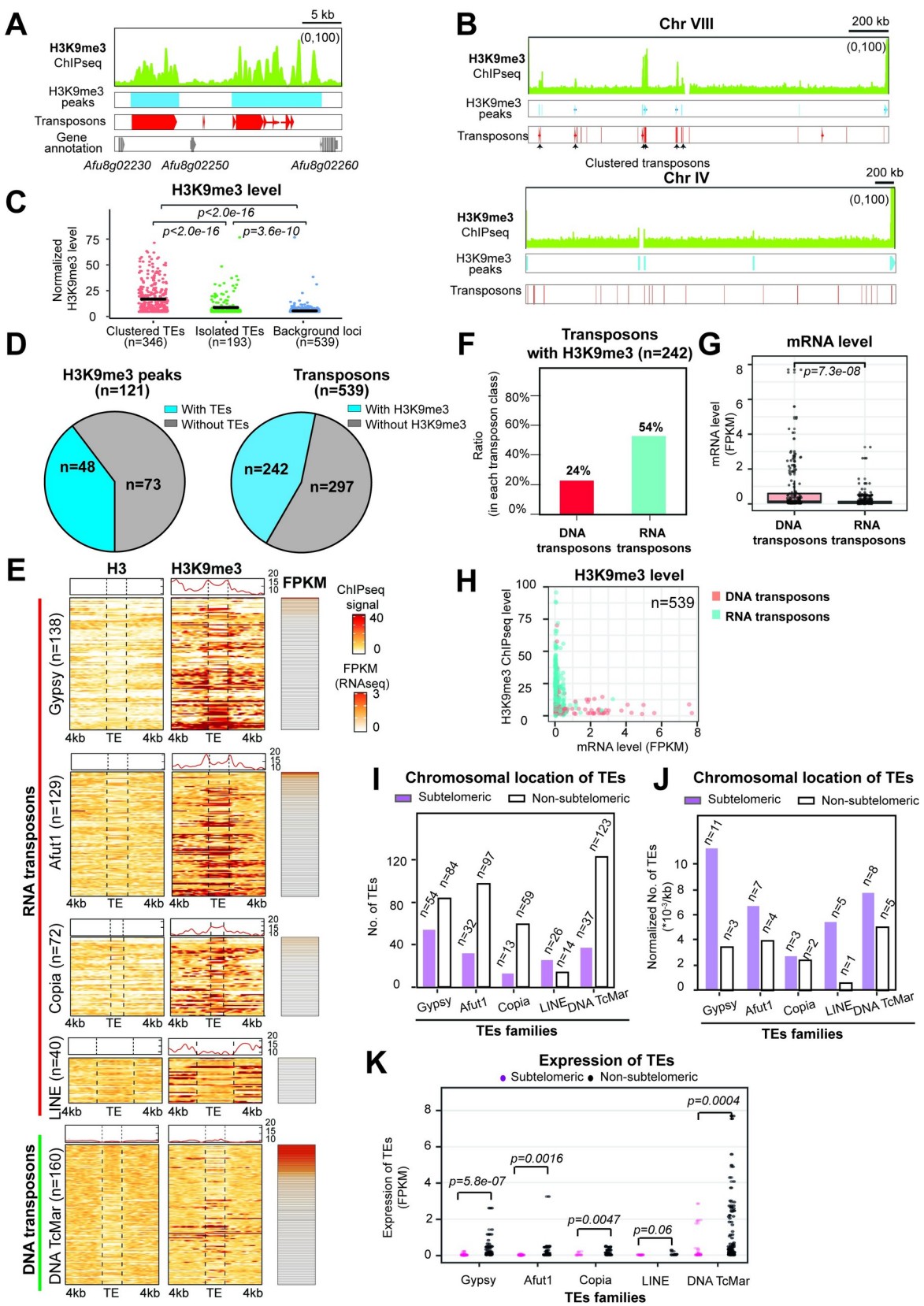

**Fig 4. Different TEs distribution between Af293 and CEA17.** (A-B) Genome browser screenshots showing the distribution of H3K9me3 peaks called by MACS2 and transposons detected by RepeatMasker in Af293. (C) A scatter plot showing the normalized H3K9me3 level at each TE location within +-25bp flanking window deposited at genome. The lines within each scatter populations represent the median value. H3 signal at each TE loci was used for normalization. Random genomic loci (n = 539) with consistent nucleotide length as TEs were selected from the TE-excluded genome and ploted as control. (D) Pie charts showing the distribution of H3K9me3 peaks with or without TE (left panel), and transposons with or without H3K9me3 (right panel) genome wide in Af293. (E) Heatmap plots showing the deposition levels of H3 and H3K9me3 at the Af293 TEs loci identified by RepeatMasker and the 4 kb boundary regions at both sides, and the expression levels as shown by FPKM. (F) A bar plot showing the number of DNA and RNA transposons bound by H3K9me3 in Af293. (G) A bar plot showing the expression levels of DNA and RNA transposons in Af293. (H) A scatter plot showing the relationship between expression level and H3K9me3 modification level at DNA and RNA transposons in Af293. (I) A diagram plot showing the number of different transposon families in Af293 at chromosomal locations (subtelomeric *vs.* non-subtelomeric regions). (J) A diagram plot showing the normalized number TEs at chromosomal locations (subtelomeric *vs.* non-subtelomeric regions) as shown in (I) when taken into account the size of two genomic regions. The TE number was normalized to the size of subtelomeric or non-subtelomeric region as shown in y-axis. (K) A scatter plot showing the expression level of different transposon families same as (I) in Af293.

To determine which class of TEs are modified by H3K9me3 in *A. fumigatus*, we used the RepeatMasker software to further sub-classify the identified TE elements. Of the 539 Af293 transposon ORFs detected, 379 were classified as RNA transposons and 160 as DNA transposons (Fig 4E). Among the RNA transposons, 138 were assigned to the Gypsy group, 129 to the Afut1 group, 72 to the Copia group and 40 to the LINE group, while all the 160 DNA transposons belonged to the DNA TcMar group (Fig 4E). Interestingly, the modification pattern seemed to differ between different TEs; for example, Gypsy, Afut1 and Copia transposons were bound by H3K9me3 at their ORFs and/or 4 kb flanking regions at both sides while the LINE transposons were bound by H3K9me3 mainly at the flanking regions but not within the transposon body (Fig 4E). It is also noteworthy that there was a distinct difference between DNA and RNA transposons in that more than half (54%) of RNA transposons were modified by H3K9me3, while only a quarter (24%) of DNA transposons had the histone mark (Fig 4F). Moreover, there were a significantly larger number of DNA transposons being expressed (n = 35) as compared to RNA transposons (n = 7) (Fig 4E and 4G), and those expressed DNA transposons were not detectably marked by H3K9me3 (*i.e.*, have very low levels of H3K9me3 ChIP-seq signal) (Fig 4H). A similar situation was observed for CEA17, which carries 654 transposons with 476 RNA transposons (133 Gypsy, 209 Afut1, 7 LINE and 72 Copia) and 178 DNA transposons (DNA TcMar) (S4B–S4F Fig). These observations indicate a preferential regulation by H3K9me3 of RNA transposons over DNA transposons.

Analysis of the chromosomal position of these elements revealed that different TE classes have dissimilar subtelomeric (*e.g.*, within 300kb of chromosome ends) versus non-subtelomeric distributions with relatively more Gypsy, Afut1, Copia and DNA TcMar TEs located at non-subtelomeric regions, while LINE TEs are mostly found at subtelomeric regions (Fig 4I and S4 Table). When the size of the two genomic regions is taken into account, TEs are found at a relatively higher frequency at the subtelomeric regions (Fig 4J). Notably, the DNA TcMar, Gypsy and Afut1 TEs located at non-sub telomeric regions had relatively higher expression levels than those situated at sub telomeric regions based on the RNA-seq data, while the LINE and Copia transposons were relatively lowly expressed regardless of genomic location (Fig 4K). These results suggest a correlation between chromosomal location, H3K9me3 deposition and DNA transposon activity in Af293.

## Af293 and CEA17 genomes have different numbers of LINE TEs

Notably, despite the fact that CEA17 (n = 654) has more TEs than Af293 (n = 539) as determined by RepeatMasker, the Af293 genome contained significantly more (*e.g.*, >5 times) LINE transposons when compared to CEA17 (S4G Fig). For a comparison, there was less than

2-fold difference in the numbers for all the other TE classes. The absence of these Af293-unique LINE transposon regions in the CEA17 genome was further confirmed by mapping CEA17 genomic DNA sequencing data to Af293 reference genome (S4H Fig) and by real time PCR analysis on the genomic DNA of CEA17 and Af293 (Fig 5A) using a pair of primers specific for a LINE-transposon coding ORF (Afu8g06290) that is not present in CEA17 based on BLAST analysis (S4I Fig).

Notably, we observed a remarkable difference in the number of mutations among the LINE TEs of Af293 and CEA17 as revealed by comparisons to the consensus DNA sequence LINE TE. The LINE-transposons in the Af293 genome accumulated less mutation as compared to CEA17 (Fig 5B and 5C), indicating potentially functional transposon activity from some LINE transposons in Af293 if expressed. The type of mutations also differ between the two isolates with the majority of the mutations (~95%) in the LINE transposons in CEA17 being C:G to T: A (Fig 5D and S5 Table), which indicates transposon inactivation by the repeat-induced point mutation (RIP) process [71], while this was not the case for Af293. No difference was observed for the DNA TE Aft1 sequence (S4J Fig) and rRNA (S4K Fig) between the two isolates. More importantly, while all the LINE transposons in CEA17 were modified by H3K9me3 (Fig 5E), only less than 1/3 of LINE transposons in Af293 had the histone mark (Figs 4E and 5E) and the modification patterns differ between CEA17 and Af293 (*e.g.*, on top of TEs versus at the flanking regions of TEs, respectively, Figs 4E and S4B), suggesting differential regulation of LINE transposons in the two isolates.

## Evidence for Af293 having a higher level of chromosomal rearrangement events

Regions enriched in TEs are often found near chromosome rearrangement events (reviewed by [72]). In light of the above TE observations, we next set out to compare macro-chromosomal changes such as chromosomal loss, if any, between Af293 and CEA17. To assess this, we took advantage of the published RNA-seq data for the Af293 (n = 83) and CEA17 (n = 114) isolates from different laboratories to identify such changes. The rationale behind this approach is that if a strain underwent a chromosomal loss, the genes residing at the lost region would have no RNA-seq reads (or very low levels due to potential background reads from non-specific mapping). Hence, a significant continuous stretch of seemingly non-expressed (or low-expressed) genes in a given strain may indicate a chromosomal loss.

Roughly an equal number of public RNA-seq datasets were processed and analysed for the two isolates (83 and 114 datasets from 19 and 28 studies by 16 and 17 laboratories for Af293 and CEA17, respectively) (S3 Data). Both isolates have largely similar datasets covering a range of experimental conditions such as defined media, mice infection, interactions with mammalian and bacterial cells (S3 Data). None of the CEA17 datasets showed noticeable signs of major chromosomal changes (S5 Fig), based on the analysis of eleven contigs including the four largest assemblies of the incomplete CEA17 genome sequence. In contrast, obvious chromosomal differences were noted in a few Af293 datasets in which some segments of a few chromosomes appeared to be missing (Figs 6A and S5). The potential missing regions range from 35 to 320 kb in sizes. In particular, a region of approximately 320 kb at the right end of the Chr VIII appeared to be lost in the Af293 strains used in three different studies (Fig 6A and 6B). These strains also lost other chromosomal regions (*e.g.*, Chr I, V or VI) (Fig 6A), and these missing regions are not the same among the isolates. Interestingly, besides this 320 kb at Chr VIII region, another region of about 100 kb in length overlapping the 320 kb region was missing in an Af293 strain used in another study (Fig 6A). Overall, the results suggest that the Af293 strains in different laboratories had independently acquired chromosomal losses with

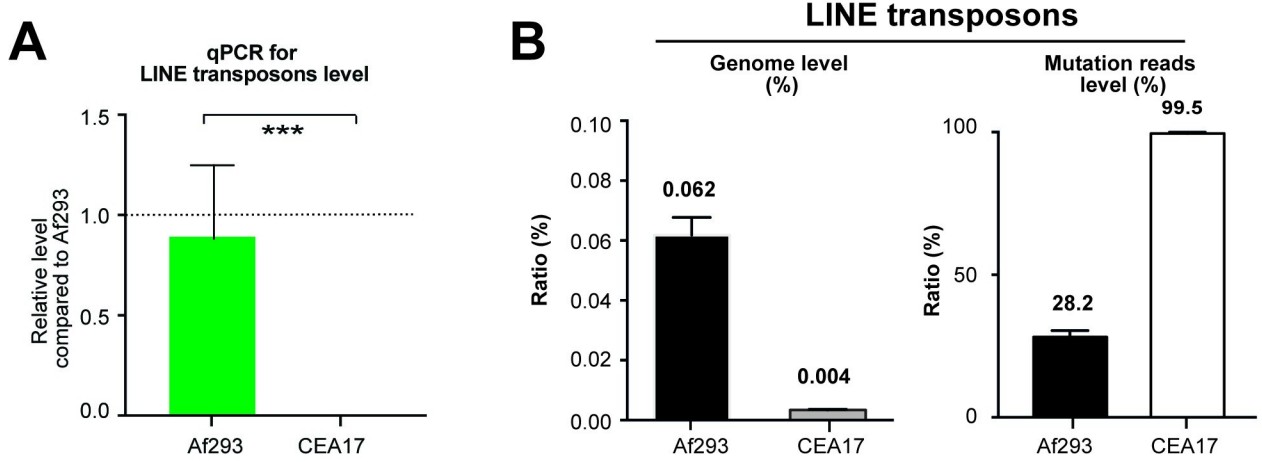

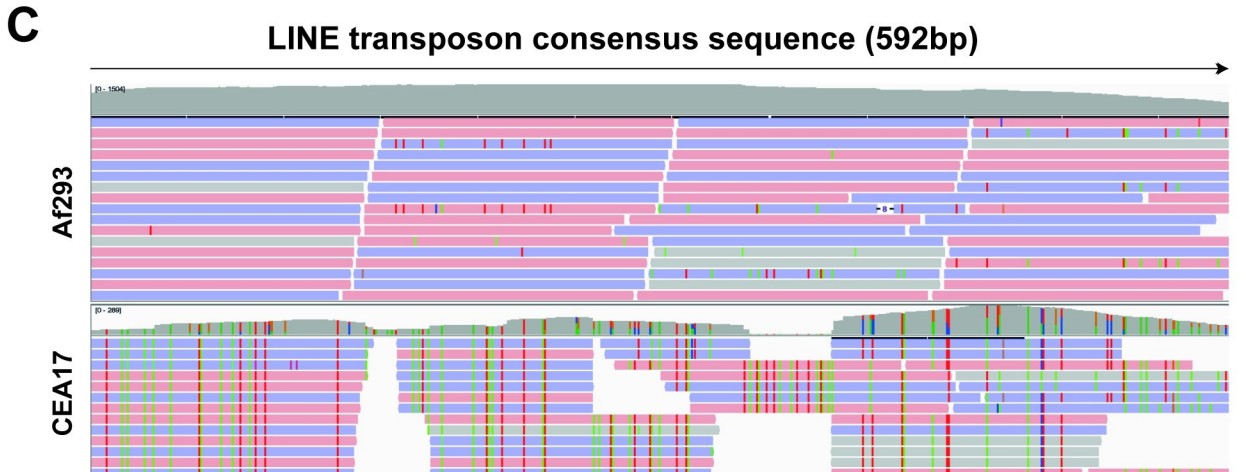

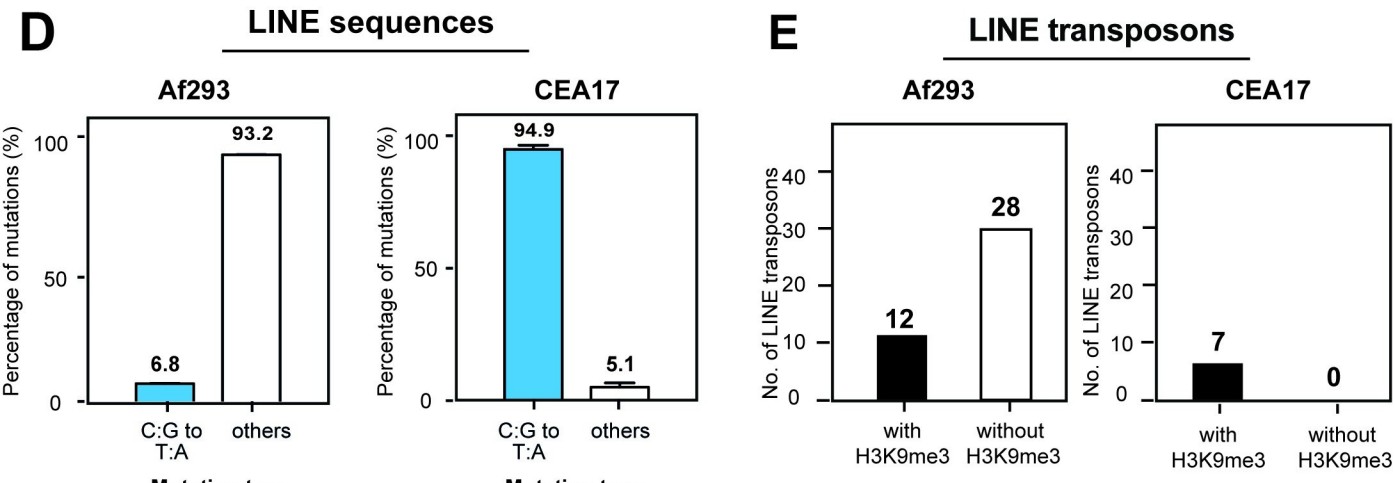

**Fig 5. Distinct distribution of LINE TEs in Af293 and CEA17.** (A) A bar plot showing the quantification of the genome level of LINE transposons in Af293 and CEA17 detected by qPCR. The qPCR was processed in two independent replicates and P value was calculated by unpaired t test. Error bar represent standard derivation and * means P value <0.05; ** means P value <0.01; *** means P value <0.005. (B) Bar plots showing the genome percentage of LINE transposons in the Af293 and CEA17 genomes and their mutation reads percentage by alignment to the LINE conserved sequence in S4 Data. Error bar represent standard derivation. (C) A genome browser screenshot showing the alignment of Af293 and CEA17 genomic DNA to a conserved LINE transposon sequence in S4 Data. The mismatched nucleotides were colorized. (D) Bar plots showing the percentage of C:G to T:A mutations in Af293 and CEA17 within those mutation reads mapped to LINE consensus sequence in S4 Data. Error bar represent standard derivation of two biological replicates. (E) Bar plots showing the number of LINE transposons with and without H3K9me3 at the TE loci or within the boundaries in Af293 and CEA17.

the Chr VIII region affected in multiple independent strains, suggesting that this region is prone to chromosomal loss (*i.e.*, a chromosomal loss hotspot).

To further demonstrate that Af293 strains in different laboratories have genetic modifications and to confirm that the lack of RNA-seq reads in the above observation was indeed due to the loss of chromosomal regions, we sequenced the DNA of an existing Af293 culture in our laboratory (referred to as Af293GG hereafter) and a newly acquired Af293 culture from the ATCC fungal collection (referred to as Af293ATCC hereafter). Interestingly, the 320 kb of the right arm of Chr VIII was also missing in our laboratory strain (Figs 6C and S6A), which was further confirmed using real time qPCR analysis (Fig 6D). It is noteworthy that this is the only missing region in the Af293GG strain when compared to Af293ATCC, in contrast to the above-mentioned strains that have lost multiple regions (the right two columns in Fig 6A), indicating that Af293GG is not related to those strains (*i.e.*, not a derivative strain or a shared strain from other laboratories).

It was noted that the chromosome loss hotspot in Chr VIII is adjacent to a LINE TE (Afu8g06290) (Fig 6B) and that expression of many LINE TEs (Fig 6E) was elevated in some of the datasets that lost the Chr VIII region (Fig 6F), raising a possibility that LINE TEs may be linked to the chromosomal loss. However, the relationship between the Chr VIII loss and these LINE TEs expressions was not strictly correlated in all datasets missing the same Chr VIII region, and, therefore, additional experiments will be needed to confirm the relationship. Nevertheless, the overall results suggest that Af293 strains are more prone to chromosomal rearrangements compared to CEA17.

## The Af293GG isolate has gained amplification of the fumitremorgin SM cluster genes and elevated production of fumitremorgin and various SMs

Genome analysis comparison revealed that the missing ~320 kb region of Chr VIII in Af293GG encompasses 109 genes including seven that encode DNA-binding transcription factors (S6 Table). Although most of these missing genes were not functionally characterized (S6 Table), GO analysis found significant enrichments for maltose metabolic process, polyamine biosynthesis and cell wall modification process (S7 Table). To determine whether the chromosomal loss in Af293GG resulted in altered gene expression, we performed transcription profiling using ChIP-seq against RNA polymerase II [73] for the Af293ATCC and Af293GG strains. Differentially expressed genes (DEGs) analysis identified 197 up-regulated and 56 down-regulated genes in Af293GG comparing to the Af293ATCC strain (S8 Table). GO enrichment analysis showed that the down-regulated genes were enriched for plasma membrane organization, thiamine and protein transport processes (S9 Table), while the up-regulated genes showed enrichment for genes involved in stress response, carbon metabolism, reproduction, pigmentation and melanin biosynthetic process and secondary metabolism (S6B Fig and S9 Table).

Consistent with the GO enrichment of pigmentation, melanin biosynthesis and secondary metabolism in the up-regulated genes of Af293GG, the mycelia mass and culture media of Af293GG (but not Af293ATCC and CEA17) were pinkish in colour (S6C Fig), indicating of

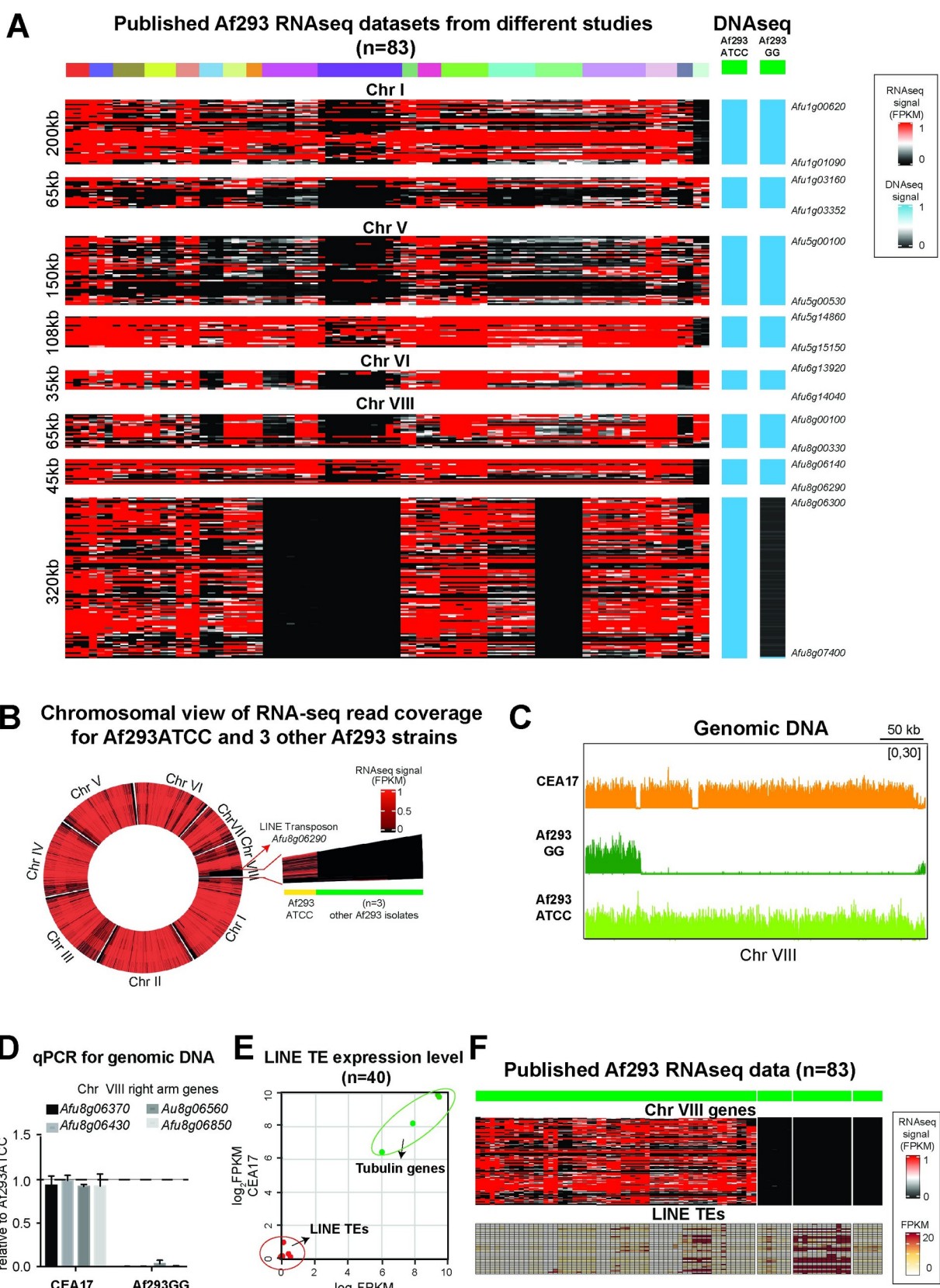

**A** Published Af293 RNAseq datasets from different studies (n=83) — DNAseq Af293 ATCC, Af293 GG

**B** Chromosomal view of RNA-seq read coverage for Af293ATCC and 3 other Af293 strains

**C** Genomic DNA

**D** qPCR for genomic DNA

**E** LINE TE expression level (n=40)

**F** Published Af293 RNAseq data (n=83)

**Fig 6. The genome instability of Af293 isolates.** (A) Heatmap plots showing selected chromosome regions from 83 published Af293 RNA-seq data and genomic DNA of the Af293 isolate from our laboratory (Af293GG) and the stock center (Af293ATCC). (B) A circle heatmap plot showing the chromosomal view of selected three RNAseq data sets with the missing region in the Chr VIII right arm. A LINE TE adjacent to the region was marked by arrow. RNAseq reads of Af293ATCC was plotted as a control. (C) A genome browser screenshot showing the genomic DNA levels of Chr VIII right arm of CEA17, Af293GG and Af293ATCC strains. (D) A bar plot showing the levels of selected genes from the right arm of Chr VIII in CEA17 and Af293GG isolate when compared to Af293ATCC. The qPCR was processed in two independent replicates and P value was calculated by unpaired t test. Error bar represent standard derivation and * means P value <0.05; ** means P value <0.01; *** means P value <0.005. (E) A scatter plot showing the expression level of LINE TEs in Af293 and CEA17. The expression of tubulin genes was plotted as a control. (F) Heatmap plots showing the LINE expression level in published 83 RNA-seq data.

different SM production by Af293GG. Therefore, we assayed for the production of various known secondary metabolites by the two strains using LC-MS. Consistent with the transcription profile showing increased expression of pyripyropene A biosynthesis genes (BGC27) in Af293GG (Fig 7A), a higher level of pyripyropene A was produced by Af293GG comparatively to Af293ATCC (Fig 7B and 7C). In addition, a few other SMs (*e.g.*, fumitremorgin C, fumigaclavine A and gliotoxin) (Fig 7D) were also produced at significantly elevated levels by Af293GG. The detection of fumitremorgin C was unexpected, as it was reported that the Af293 isolate is incapable of producing fumitremorgin due to a point mutation (R202L) in the *ftmD* gene that compromises the methyltransferase activity by 20 fold [74]. The DNA sequence of the *ftmD* gene in the Af293GG variant confirmed the presence of the R202L point mutation (S6D Fig). Interestingly, in the course of inspecting the DNA sequencing data on the genomebrowser, it was noticed that the Af293GG variant might have amplified a region of 55 kb on the Chr VIII left arm adjacent to a LINE TE Afu8g00310 (Fig 7E). The amplification was further corroborated by real time qPCR using primer pairs specific to three different loci within the region (S6E Fig). The quantitative PCR result showed a 2-fold increase in the DNA level, suggesting a duplication of this region. The duplication encompasses 20 genes with eleven being uncharacterized and nine genes belonging to the fumitremorgin A-C biosynthetic cluster (S10 Table). The amplification of the cluster was further confirmed by quantitative PCR on three of the cluster genes [*fmtC* (Afu8g00190), *fmtD* (Afu8g00200) and *fmtE* (Afu8g00220)] (Fig 7F) and might have consequently increased the expression of these cluster genes including the *ftmD*$^{R202L}$ mutant gene, thereby increasing fumitremorgins production in Af293GG. Indeed, RT-qPCR detected 5-to-10-fold higher mRNA levels on three of the cluster genes (*ftmC*, *ftmD* and *ftmE*) in Af293GG when compared to Af293ATCC, and their levels were even higher than those in CEA17 (Fig 7G). However, there was no difference in H3K9me3 modification between Af293GG and Af293ATCC at the fumitremorgin cluster (Fig 7H) as well as all other SM clusters (Fig 7I), which is in agreement with the lack of association between BGCs regulation and H3K9me3 modification described above.

Fumitremorgins are indole alkaloids, whose biosynthesis begins by condensation of the two amino acids L-tryptophan and L-proline to brevianamide F, catalyzed by FtmA. Brevianamide F is then prenylated by FtmPT1/FtmB, resulting in the formation of tryprostatin B. FtmE is responsible for the conversion of tryprostatin B to demethoxyfumitremorgin C and tryprostatin A to fumitremorgin C, while the conversion of tryprostatin B to 6-hydroxytryprostatin B and 6-hydroxytriprostatin B to tryprostatin A are catalyzed by FtmC and FtmD, respectively. The subsequent reactions result in the formation of 12,13-dihydroxyfumitremorgin C, fumitremorgin B, verruculogen and fumitremorgin A. Consistent with the biosynthetic pathway, LC-MS analysis revealed that all the intermediates in the fumitremorgin biosynthetic pathway (S7A Fig) were produced in higher amounts by Af293GG (Figs 7J and S7B). Moreover, there was a clear distinction in the levels of intermediate metabolites before and after the biosynthesis step mediated by FtmD$^{R202L}$: the intermediates brevianamide F, tryprostatin B and 6-hydroxytriprostatin B, which are produced before the step limited by the mutated enzyme

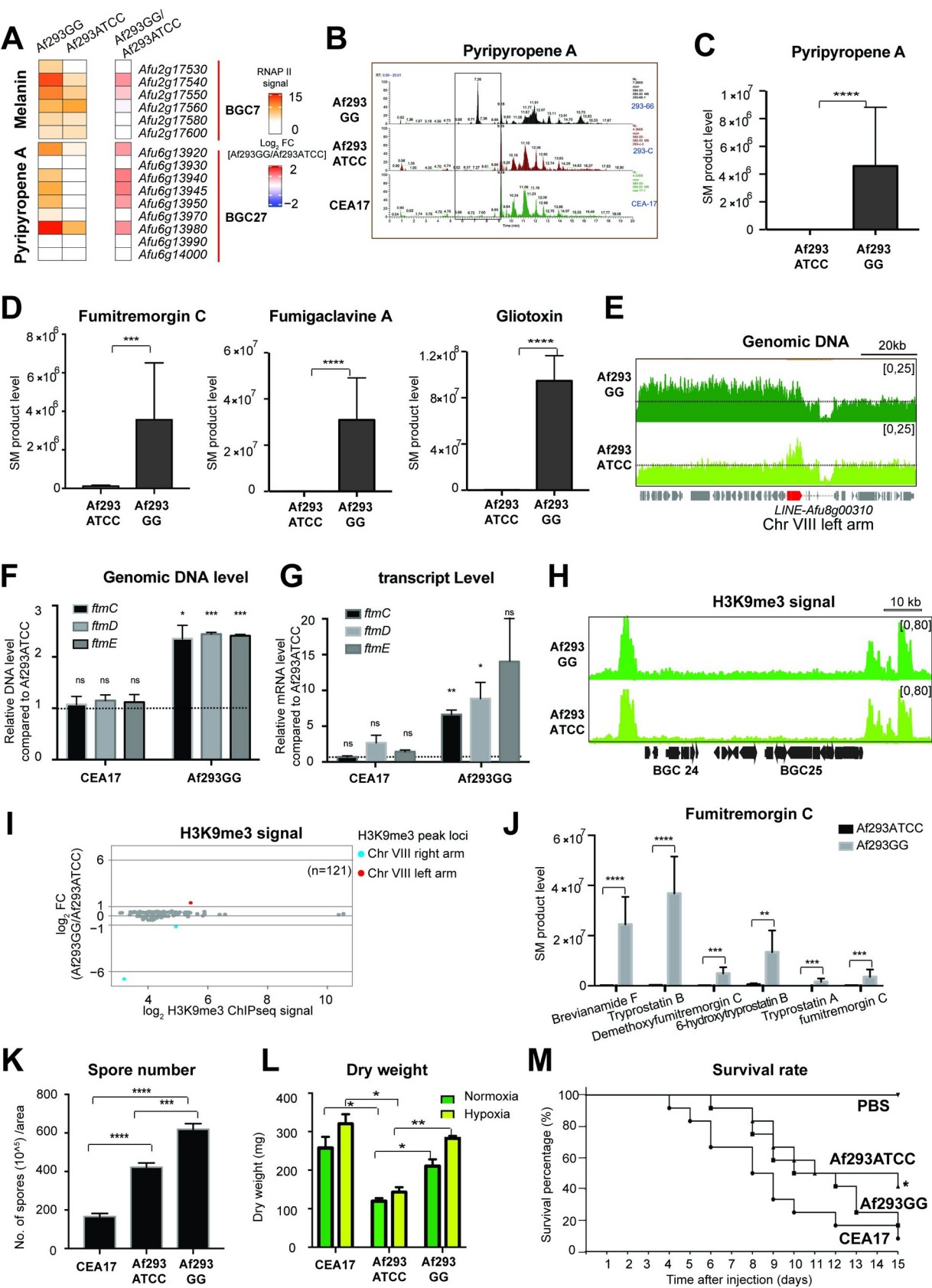

**Fig 7. The evolved Af293GG isolate induced production of many SM products and gained amplification of Chr VIII left arm, better growth fitness and higher virulence.** (A) Box plots showing the expression level detected by RNAP II ChIPseq of melanin and pyripyropene A biosynthetic genes in Af293GG and Af293ATCC. (B) LC/MS profiles showing the peaks representing pyripyropene A products in Af293GG, Af293ATCC and CEA17. (C-D), Bar plots representing the quantified production of SMs (C) pyripyropene A as shown in (B) and fumitremorgin C, fumigaclavine A and gliotoxin (D) in Af293ATCC and Af293GG. The SMs levels were calculated as peak area, and P values were calculated by F test to compare variations. Error bar represent standard derivation and * means P value <0.05; ** means P value <0.01; *** means P value <0.001, **** means P value <0.0001. (E) A genome browser screenshot showing the genomic DNA level of Chr VIII left arm of CEA17, Af293GG and Af293ATCC. The transposon ORF Afu8g00310 is marked in red. (F) A bar plot showing the relative genomic DNA level detected by qPCR of selected coding (*fmtC*, *ftmD* and *ftmE)* regions of Chr VIII left arm in CEA17 and Af293GG genomes when compared to Af293ATCC. The qPCRs were processed in two independent replicates and P values were calculated by unpaired t test. Error bar represent standard derivation and * means P value <0.05; ** means P value <0.01; *** means P value <0.005. (G) A bar plot showing the qPCR relative quantification of the expression levels of *fmtC-E* genes in CEA17 and Af293GG isolates when compared to Af293ATCC. The qPCRs were processed in two independent replicates and P values were calculated by unpaired t test. Error bar represent standard derivation and * means P value <0.05; ** means P value <0.01; *** means P value <0.005. (H) A genome browser screenshot showing the H3K9me3 deposition at the boundaries of BGC 24 and BGC 25 in Af293ATCC and Af293GG. (I) A MA plot representing the relative H3K9me3 ChIP-seq signal in Af293GG when compared to Af293ATCC. The red dot is the peak located at the Chr VIII left arm, and the two blue dots are the peaks located at the Chr VIII right arm. (J) A bar plot showing the quantified production of fumitremorgin C and its intermediates in Af293GG and Af293ATCC as shown in S7B Fig, Error bar represent standard derivation and * means P value <0.05; ** means P value <0.01; *** means P value <0.001, **** means P value <0.0001 as shown by F test to compare variations. (K) A bar plot showing the number of spores per area produced by CEA17, Af293ATCC and Af293GG strains. (L) A histogram plot showing the dry weight of CEA17, Af293ATCC and Af293GG biomass when the strains were grown in static liquid cultures exposed to normoxia (~21% $O_2$) and hypoxia (0.9% $O_2$) conditions for 96 hours. Error bars in k-l represent standard derivation and * means P value <0.05; ** means P value <0.01; *** means P value <0.001, **** means P value <0.0001 as shown by t-test to compare variations. (M) Survival curves showing the survival percentage of *G. mellonella* (n = 12/strain) infected with CEA17, Af293GG and Af293ATCC spores. Phosphate buffered saline (PBS) without conidia was used as a negative control. Indicated P-values are based on the Log-rank, Mantel-Cox and Gehan-Breslow-Wilcoxon tests. * means P value <0.05.

were accumulated at higher levels comparing to the intermediates tryprostatin A and fumitremorgin C, which are produced in the subsequent steps (Fig 7J). This observation is consistent with the reduced methyltransferase activity of FtmD$^{R202L}$, which imposes a rate-limiting step in fumitremorgin biosynthesis. The increased *fmtD*$^{R202L}$ expression, presumably together with elevated levels of precursor intermediate metabolites, could augment the compromised methyltransferase activity of FtmD$^{R202L}$ for fumitremorgin production. Alternatively, another methyltransferase could be acting in this pathway. Taken together, these results demonstrated that our evolved Af293GG strain has acquired enhanced abilities for SM production as a result of major chromosomal changes (*i.e.*, amplification and loss).

## The Af293GG isolate has increased asexual spore production, growth and virulence

Growth analysis showed that Af293GG and Af293ATCC as well as CEA17 have slightly different colony morphology (S8A Fig) and colony colour, which may indicate a variation in spore production. Indeed, the three strains produced different numbers of asexual spores under the same conditions with Af293GG having the most (Fig 7K). Moreover, Af293GG also generated higher biomass than Af293ATCC both in normoxia and hypoxia growth conditions at levels similar to CEA17 (Fig 7L). Given the positive effects of hypoxia fitness [55] and mycotoxin production (*e.g.*, gliotoxin) towards *A. fumigatus* virulence, we next determined whether the evolved Af293GG isolate also has increased virulence using the *Galleria mellonella* larvae infection model. Larvae infected by CEA17 conidia began to die on the 4th day, while the larvae infected by Af293ATCC and Af293GG conidia started dying on the 6th day (Fig 7M). After 11 days, the larvae infected by Af293GG conidia were more melanised and less active than the ones infected by Af293ATCC (S1 Movie). On the 15th day, 8%, 40% and 16% of the larvae infected by CEA17, Af293ATCC and Af293GG strains remained alive (Fig 7M) with a median survival day of 8.5, 13 and 11 days, respectively (S8B Fig). Although all three strains were able to cause death eventually, there are noticeably differences in those measured infection attributes between the laboratory evolved Af293GG strain and the standard Af293ATCC isolate.

## Discussion

In this work, we performed the first *A. fumigatus* chromatin profiling for histone H3 and two histone modifications, H3K4me3 and H3K9me3, to identify the similarities and differences between the two most commonly studied clinical isolates Af293 and CEA17. We showed that H3 depletion at core promoters and H3K4me3 modification at 5' end of transcribed genes positively correlate with transcriptional activity, which is consistent with what has been observed in other eukaryotes [23,75]. It is noteworthy that the functional significance of H3K4me3 in transcription is not clear; i.e., while this chromatin modification has been largely associated with active transcription, the lack of this modification has little effect on transcription in both yeast and mammalian cells [23]. Nevertheless, existing evidence indicates H3K4me3 in the recruitments of chromatin remodelers and other proteins to modulate RNAP II transcription and/or its associated processes. Therefore, H3K4me3 is generally taken as a histone mark for transcribed genes [23,76]. Our results show the same situation in *A. fumigatus*, as expressed genes such as those participating on primary metabolism and cellular component organization and biosynthesis are marked by H3K4me3 in both isolates, while most secondary metabolism genes, which are silent, are devoid of this histone modification. Unexpectedly, a group of well-expressed genes (based on the presence of mRNAs) including many transporter genes were not marked by this modification. One explanation for this is that the transcription of those genes does not involve active histone modification as shown for mammalian developmental genes [77]. Alternatively, those genes are controlled by a unique chromatin regulatory mechanism that involves other histone modifications but not H3K4me3 [78]. Yet another non-mutually exclusive explanation is related to the fact that ChIP-seq experiments, through formaldehyde crosslinking, capture events that occur at the time of cell harvest. Therefore, those transporter genes may be transcribed earlier before the actual experiment (*i.e.*, during spore germination or the initial growth stage) but their mRNAs are stable enough until the point of experiment for detection by RNA-seq, whereas H3K4me3 was removed soon after transcription ceased. Accordingly, the finding from a transcriptome analysis of germinated spores [79] supports the second hypothesis, showing that transporter genes (*e.g.* ABC drug efflux pumps (*abcB* and *abcD*) and MFS (*msfB* and *msfC*) are induced at early germination stages (30 minutes) and their expressions cease at later time points (90 minutes). If this last explanation were indeed true, our H3K4me3 data could offer a simple mean to demarcate early (e.g., during germination) versus recent (e.g., during active hyphal growth) gene expression events when analysed with RNA-seq data.

The histone modification H3K9me3, which is usually associated with heterochromatin [24], was also systematically investigated. Here, we showed that H3K9me3 are found at telomeric, sub-telomeric and pericentromeric regions, similar to what had been reported in other eukaryotes such as *Schizosaccharomyces pombe* [80,81] and *N. crassa* [82,83] but unlike *C. albicans* [84,85] and *Saccharomyces cerevisiae* [86]. The (sub)telomeric localization has been previously associated with transcriptional regulation of SM BGCs (see [87] for a review). However, we found here that most of *A. fumigatus* SM BGCs are not marked by H3K9me3 at the genome-wide level and that the modification does not strictly represent BGC silencing. In other words, our observations suggest a minor role of H3K9me3 on SM BGC regulation. This raises an interesting question as to how is silencing of SM BGC achieved in *A. fumigatus*, given that the other major heterochromatic mechanism mediated by Polycomb Repressive Complex and H3K27 methylation is also not present [88].

Both Af293 and CEA17 genomes contain higher numbers of retrotransposons than DNA transposons, and we showed that the retrotransposons were more frequently associated with H3K9me3. It was also noted from the RNA-seq data that DNA transposons appears to be

relatively more expressed than retrotransposons, especially those at non-subtelomeric regions, suggesting that both chromosomal location and H3K9me3 deposition could influence TE expression and perhaps even TE activity in *A. fumigatus*. The suppressive role of H3K9me3 on transposons has been experimentally demonstrated in other eukaryotes such as *Zymoseptoria tritici* [89] and *C. elegans* [90], where loss of H3K9me3 lead to increased TE expression, although a recent study in the fungal plant pathogen *Verticillium dahliae* showed that different transposon families may have distinct chromatin modification profiles [91]. Our result that H3K9me3 does not uniformly mark different classes of TEs indicates a similar situation in *A. fumigatus*.

Another interesting finding is that Af293 contains significantly more LINE-transposons than the CEA17 isolate, which was also reported in a previous study [92]. Our analysis revealed that a high percentage (~70%) of LINE-transposons in Af293 were not subjected to H3K9me3 modification (Fig 5E) and that the LINE transposon sequences in Af293 contain relatively much less mutations than in those in CEA17 (Fig 5B and 5C). Based on these, it is tempting to speculate higher transposon activities in the Af293 isolate. In support of the speculation and consistent with the association of transposons activity with chromosome rearrangement and genome instability in different organisms [89,90,93–96], we indeed detected gross chromosome alteration events including a chromosomal loss hotspot in Chr VIII in Af293 strains of different laboratories but not from different CEA17 strains. This circumstantial evidence leads us to a proposal that the higher number of LINE transposons in Af293 may be linked to the increased incidents of chromosomal rearrangements observed among Af293 strains. It will be interesting to experimentally confirm this (*e.g.*, by microevolution experiments in the absence of H3K9me3 modification and/or with increased transposon activity) and to compare transposon numbers and activities between different clinical isolates, considering chromosomal rearrangement has been shown as a virulence mechanism for the adaptation of pathogenic *Candida* species to different host niche environments [47,97].

Fungi utilize a number of genome defense mechanisms for protection against TEs including RIP that was first identified in *N. crassa* [98]. Although RIP has not been demonstrated in *A. fumigatus*, an early study had implicated its existence and protection against TEs [99]. It is noteworthy that the mutations observed at the LINE transposons in CEA17 (*e.g.*, >95% mutations are C:G to T:A transitions) is highly indicative of transposon inactivation by RIP [71]. Remarkably, the Af293 isolate has a strikingly contrasting mutational profile in which only around 7% of the mutations are C:G to T:A transitions. Consistent with RIP functioning in conjunction with H3K9me3 heterochromatic silencing [100], H3K9me3 modification was also found on top of all CEA17 LINE transposons but not on Af293 LINE transposons. If RIP indeed operates in *A. fumigatus*, the interesting heterogeneities in their LINE mutational profiles potentially suggests different levels of genome defence by RIP between the two isolates. In *N. crassa*, the RIP process occurs during the sexual stage, and if it is also conserved in *A. fumigatus*, the result may suggest another mutually non-exclusive possibility that the two isolates have different levels of sexual activities. Alternatively, the C:G to T:A transitions may be due to a DNA methyltransferase-mediated mechanism that occurs during mitosis as found for the plant fungal pathogen *Z. tritici* [101]. Regardless of the mechanism, our result indicates differential TE inactivation activities between *A. fumigatus* clinical isolates.

Beyond the heterogeneities in H3K9me3 modification and TE profiles, this work also reports unexpected genetic heterogeneities among Af293 strains used by different laboratories. A number of chromosomal alterations were observed with one region at Chr VIII (including seven transcription factor encoding genes and genes that codify proteins involved in cell wall metabolism) lost in different variants, implying that this region is a rearrangement hotspot in Af293. In addition, the Af293GG genome had a chromosomal amplification on the other end of the same chromosome that contains the BGC genes for fumitremorgin. Consistent with the

amplification, an increased production of fumitremorgin C was detected. The *ftmD* gene, in the fumitremorgin cluster, encodes an O-methyltransferase that catalyzes the conversion of 6-hydroxytryprostatin B into tryprostatin A [74]. The Af293 has a R202L mutation that blocks fumitremorgin B synthesis in this strain [74]. Since the same mutation is detected in Af293GG *fmtD* gene, the much higher and detectable production of fumitremorgin C and its intermediates in Af293GG strain could be due to the augmented FmtD residual activity caused by its overexpression or another *O*-methyltransferase that functions at this biosynthetic step. Interestingly, several other SMs such as fumigaclavine A, pyripyropene A and gliotoxin were also produced in higher levels in the Af293GG variant than the original Af293 isolate. The reason for these increases is unclear and could be due to a number of plausible mechanisms; *e.g.*, misregulation of the SM BGCs or altered physiology due to the lost of the Chr VIII right arm that includes many transcription factor genes and to the metabolic imbalance caused by the fumitremorgin over-production, *etc*. Although most BGCs contain specific transcription factors involved in the transcriptional regulation of determined SM clusters, an increasing number of evidence have demonstrated cross-regulation between the different BGCs [102–105]. For instance, an iron dependent network composed by SreA and HapX induces the hexadehydroasthechrome (HAS) BGC under iron excess [104]. The *hasA* overexpression, in turn, results in downregulation of several genes involved in pyripyropene, fumitremorgin and gliotoxin biosynthesis [104]. In fact, gliotoxin and fumitremorgin regulation mechanisms are probably related, as the disruption of dithiol gliotoxin *bis*-thiomethylation by *gmtA* deletion results in decreased gliotoxin, tryprostatin B and fumitremorgin production in *A. fumigatus* [105]. Inversely, here we show gliotoxin production as a result of the fumitremorgin BGC upregulation, reinforcing the idea that fungal BGCs are controlled by diverse regulatory mechanisms and in interconnected manners [87].

The identification of the Af293GG and other Af293 variants used by different laboratories implies additional levels of complexities in the heterogeneities of *A. fumigatus*, emphasizing the fascinating ability of this important fungal pathogen to evolve new growth and fitness characteristics that could influence virulence. Our study suggests a potential link between epigenetic heterogeneity (*e.g.* H3K9me3 modification), TEs, genome instability and genetic heterogeneity. Confirming a causative relationship between these events and identifying additional mechanisms that drive the heterogeneities will guarantee insights into *A. fumigatus* virulence.

## Materials and methods

### Strains and culture conditions

Two *A. fumigatus* clinical isolates (Af293, which is obtained from ATCC and referred to as Af293ATCC, and CEA17) and one spontaneous Af293 variant (referred to as Af293GG) were used in this project. Conidia were harvested after 2 days of growth at 37°C on solid complete medium [106]. Mycelia samples were obtained by growing $5x10^{\wedge 7}$ conidia in flasks (250 ml Erlenmeyer's)) containing 100 ml liquid ANM medium [106] using sodium nitrate as the nitrogen source in an orbital shaking incubator at 220 rpm and 37°C for 16 hours. For the hypoxia assay, conidia were inoculated in flasks (125 ml Erlenmeyer's) containing 30 ml of liquid ANM medium and incubated without agitation for 96 hours at 0.9% $O_2$. The flasks incubated under normal atmospheric conditions (normoxia) were used as control.

### Chromatin preparation

Chromatin was prepared as described previously [107]. Briefly, formaldehyde was added to cultures at 1% final concentration and gently mixed for 20 minutes at room temperature. Next, a final concentration of 0.5 M glycine was added and further incubated for 10 minutes.

Crosslinked mycelia were harvest by filtering, washed with cold water, press-dried on paper towels, snap-frozen in liquid nitrogen and stored at -80˚C until use. For chromatin extraction, frozen mycelia were freeze-dried for 3–4 hours and lysed by 6 times of 3 minutes mechanical beatings in the presence of ~100 µl volume of silica beads using Bullet Blender (Next Advance) with at least 3 minutes of cooling in between each cycle. Samples were sonicated using the Qsonica Q800R at 100% amplitude with 10 seconds ON and 15 seconds OFF cycles for a total sonication time of 30 minutes. Chromatin concentration and size (100-500bp) were checked on 2% agarose gel, and the prepared chromatins were stored at -80˚C until use.

### Chromatin Immuno-precipitation and sequencing library preparation

Chromatin Immuno-precipitation was carried out as previously described [108,109] using antibodies listed in S11 Table. Immuno-precipitated materials were purified using QIAGEN PCR cleanup kit (cat no. 28106) and subjected to library preparation as described previously [110] with a minor modification in replacing the end-repair step with the use of NEBNext Ultra II End Repair/dA-Tailing module (NEB, cat. no. E7546L) accordingly to manufacturer's protocol. One ng of chromatin DNA was used for library preparation as input DNA control. Libraries were checked and quantified using DNA High Sensitivity Bioanalyzer assay (Agilent, cat. no. XF06BK50), mixed in equal molar ratio and sequenced using the Illumina HiSeq2500 platform at the Genomics and Single Cells Analysis Core facility at the University of Macau.

### ChIP-seq data mapping and analysis

Raw sequencing reads of histone ChIPseq experiments were checked for quality using FastQC (http://www.bioinformatics.babraham.ac.uk/projects/fastqc/) and aligned to *A. fumigatus* reference genomes (Af293 [genome version s03-m05-r06] or CEA17 [A1163]) using Bowtie2 (version: 2.2.9) using parameters that allow no mismatch [111]. For H3K9me3 peaks calling, MACS2 was applied using the parameter [macs2 callpeak—nomodel -t ${file} -f BAM -g 29420142 -n ${file}_macs2—broad—broad-cutoff 0.000001]. H3K9me3 peaks were identified by MACS2 analysis. H3K9me3 peaks adjacent to each other (e.g. within 500 bp) were manually combined as one peak. Genes next to or around H3K9me3 modified regions or given TEs were identified using an in-house script https://github.com/zqmiao-mzq/closest_gene_calling/ blob/master/find_closest_TE_vs_peak.pl. ChIP-seq signals were normalized to total sequence reads. All ChIP-seq experiments were performed in two biological replicates, and all reported patterns were observed in both replicates (S9A–S9B Fig). Gene Ontology (GO) enrichment analysis [112] was performed in FungiDB and AspGD. H3K4me3 and RANP II ChIP-seq signal levels at gene coding regions were measured using an in-house script https://github.com/ zqmiao-mzq/closest_gene_calling/blob/master/zqWinSGR-v4.pl.

### Genomic DNA extraction and real-time quantitative PCR (qPCR)

Genomic DNA of *A. fumigatus* was extracted from frozen mycelia as described [113]. One ng of DNA was used for real-time qPCR analysis using Premix Ex Taq DNA polymerase (Takara, cat. no. RR039W) on ABI Fast 7500 (Applied Biosystems) Real-Time PCR machine using oligos listed in S12 Table. The ΔΔCt method was used for quantification using the tubulin gene *tubA* as an internal reference.

### DNA preparation for Illumina sequencing

Genomic DNA was extracted according to [113]. Extracted DNA was sonicated to 100–500 bp using the Qsonica Q800R at 100% amplitude with 10 seconds ON and 15 seconds OFF cycles

for a total sonication time of 10 minutes. One μg of sonicated DNA was used for library preparation using Illumina NEBNext Ultra Directional DNA Library Prep Kit (NEB, cat. no. 7645) according to manufacturer's protocol. Libraries were checked and quantified using DNA High Sensitivity Bioanalyzer assay (Agilent, cat. no. XF06BK50), mixed in equal molar ratio and sequenced by the service provider Novogene in China. All experiments were carried out in two biological replicates.

## DNA-seq data mapping

Raw reads from Illumina sequencing were aligned to the *A. fumigatus* reference genome Af293 (genome version genome version: s03-m05-r06) or CEA17 [A1163] using bwa (alignment via Burrows-Wheeler transformation) (version: 0.7.17-r1188) [114].

## Genome coverage calculation

Genome size of Af293, CEA17 and the *A. fumigatus* isolates from a previous study [65] (n = 79) were estimated by summing up their chromosomal or contig assemblies. The genome size of *A. fumigatus* was determined from the average size of all isolates, and the genome coverage for each isolate was calculated by dividing their corresponding genome sizes by the average size.

## RNA extraction for reverse transcription and real-time quantitative PCR (qPCR)

Mycelia were harvested and washed with cold water followed by freezing in liquid nitrogen in the presence of 1ml TRIzol (Ambion, cat. no. 135405) before storing at -80˚C until use. Cell lysis was carried out through six cycles of 3 minutes beating in 1 ml ice-cold TRIzol with ~100 μl volume of silica beads using Bullet Blender (Next Advance). Total RNA was extracted as previously described [115] and the quality was checked on 2% agarose gel. One μg of total RNAs was subjected to reverse transcription into cDNAs using PrimeScript RT reagent Kit with gDNA Eraser (Takara, cat. no. RR047A) according to manufacturer's protocol. The resultant cDNA samples were diluted twenty folds and 2 μl of diluted samples were subjected to real-time qPCR analysis as described above.

## RNA purification and preparation for RNA-Seq

Ten μg of total RNAs was treated with DNase I (NEB, cat. no. M0303) as described and quality was checked using Agilent RNA Nano LabChip kit (Agilent Technologies, cat. no. 5067–1511) by Bioanalyzer. All RNA samples used had a RNA Integrity Number (RIN) value of at least 7.0. One μg of purified RNAs were used for library preparation using Illumina NEBNext Ultra Directional RNA Library Prep Kit (NEB, cat. no. 7420) according to manufacturer's protocol. Libraries were checked and quantified using DNA High Sensitivity Bioanalyzer assay (Agilent, cat. no. XF06BK50), mixed in equal molar ratio and sequenced using the Illumina HiSeq2500 platform at the Genomics and Single Cells Analysis Core facility at the University of Macau.

## RNA-seq data mapping and analysis

Raw reads were aligned to *A. fumigatus* Af293 reference genome (genome version: s03-m05-r06) using hisat2 (version: 2.1.0) [116] expression level (*e.g.* FPKM) for each annotated gene was calculated using StringTie (version: 1.3.3b) [116]. All RNA-seq experiments were performed in three biological replicates, which are highly correlated (S9C Fig). For

published RNA-seq data of Af293 and CEA17 background, raw reads were aligned to their respective *A. fumigatus* reference genome (s03-m05-r06 or A1163), unless otherwise indicated.

## TEs analysis using RepeatMasker

TEs were identified using RepeatMasker-4.1.1 (http://www.repeatmasker.org), which reports Repeats and TEs according to previously established TE consensus. The Af293 and CEA17 genome references were used as an input to run the program based on background repeat libraries (Dfam3.1 and Repbase) [117]. For the TE analysis on the raw genome sequencing data, reads that could not be mapped to the indicated reference genome (*i.e.*, unmapped) were used as input sequence, and the numbers of reads matching to TE repeat consensus sequence were counted.

## Mismatch ratio calculation of LINE TE families

The raw sequencing reads of Af293 and CEA17 genomic DNA were aligned to the consensus sequence of LINE transposon and Aft1 transposon with the 18S rRNA sequence used as a control (S4 Data). If the mapped read contains one or more than one mismatched nucleotide, the read is recorded as mismatched read. The mismatch reads ratio was calculated as the number of mismatched TE/rRNA reads divided by the total TE/rRNA mapped reads to Af293 genome reference.

## High-resolution mass spectrometry analysis

$1 \times 10^4$ spores of each strain were inoculated in 70 ml of liquid ANM and incubated for 72 hours at 37˚C shaking at 220 rpm. The supernatants were freeze-dried and 100 mg of the dried material were extracted with methanol in ultrasonic bath for 40 minutes. The extracts were then filtered at 0.22 μm PTFE. High-resolution mass spectrometry analyses were performed in a Thermo Scientific QExactive Hybrid Quadrupole-Orbitrap Mass Spectrometer. Analyses were performed in positive mode with *m/z* range of 100–1500; capillary voltage at 3.5 kV; source temperature at 300˚C; S-lens 50 V. The stationary phase was a Thermo Scientific Accucore C18 2.6 μm (2.1 mm x 100 mm) column. The mobile phase was 0.1% formic acid (A) and acetonitrile (B). Eluent profile (A/B %): 95/5 up to 2/98 within 10 minutes, maintaining 2/98 for 5 minutes and down to 95/5 within 1.2 minutes held for 3.8 minutes. Total run time was 20 minutes for each run and flow rate 0.2 mL/minutes. Injection volume was 5 μL. MS/MS was performed using normalized collision energy (NCE) of 20, 30 and 40 eV; maximum 5 precursors per cycle were selected. MS and MS/MS data were processed with Xcalibur software (version 3.0.63) developed by Thermo Fisher Scientific.

## Infection of *G. mellonella*

Larvae of *G. mellonella* in the final (sixth) instar larval stage of development, weighing 275–330 mg were used. Fresh 2 day conidia from each strain were harvested from ANM plates in PBS solution and filtered through a Miracloth (Calbiochem). For each strain, spores were counted using a hemocytometer and stock suspensions of $2 \times 10^8$ conidia/ml were prepared. The viability of the administered inoculum was determined by plating a serial dilution of the conidia on ANM medium at 37˚C. $1 \times 10^6$ conidia in 5 μl was injected to each larva and 5 μl of PBS was used as a control for the killing effect due to physical trauma. Injection into the last left proleg was carried out using a Hamilton syringe (7000.5KH). After infection, the larvae were

maintained in petri dishes at 37°C in the dark and their morbidity and mortality were scored daily. Larvae were considered dead by presenting the absence of movement in response to touch.

## Supporting information

**S1 Fig. The similar nucleosome depletion and H3K4me3 modification in two *A. fumigatus* isolates Af293 and CEA17.** (A) A bar plot showing the genome size of Af293, CEA17 and the sequenced *A. fumigatus* isolates described in [64,65]. (B) Line plots showing the H3 and H3K4me3 deposition within 1 kb of TSS region of genes genome wide in CEA17 when the data was mapped to Af293 and CEA17 (A1163) reference genome, respectively. Gene order was ranked by mRNA level from high to low. The pink shade was plotted as shown the mapping was performed to CEA17 genome reference. (C) Scatter plots showing the correlation of mRNA level (upper panel) and H3K4me3 deposition level (bottom panel) in Af293 and CEA17. (D) Genome browser screenshots showing the mRNA and H3K4me3 level at selected genes in CEA17. H3 was used as control. (E) Gene Ontology analysis of gene sets in cluster 1–4 as shown in Fig 1E-F.
(TIF)

**S2 Fig. The H3K9me3 modification profile in CEA17.** (A) A genome browser screenshot showing the H3K9me3 ChIPseq signals in the subtelomeric and pericentromeric regions in CEA17 isolate when mapped to Af293 genome reference. Blue shade was marked with the annotated centromere region. (B) Genome browser screenshots showing the H3K9me3 ChIPseq signals at the two ends (300 kb) of four CEA17 contigs. (C) A bar plot showing the number of H3K9me3 peaks located in pericentromeric (n = 20), subtelomeric (n = 21) and other genomic regions (n = 80). (D) Genome browser screenshots showing the H3K9me3 profile of regions with dissimilar modification between Af293 and CEA17. Arrows mark the loci with different (green and orange) H3K9me3 depositions in two isolates. The pink shade in (B) and (D) represents the mapping was performed to CEA17 genome reference.
(TIF)

**S3 Fig. Majority of BGCs in *A. fumigatus* were not marked by H3K9me3 in Af293 and CEA17.** (A) A scheme diagram showing the inverted distribution of BGCS 9–16 in Af293 Chr III and CEA17 Scf-03. (B) Genome browser screenshots showing the selected BGCs [BGC 12, 16, 21 and 33] marked with H3K9me3 in Af293 [BGC 12, 16 and 21] or CEA17 [BGC 33]. TEs were indicated with purple markers in (A) and (B). (C) Bar plots showing the span length of H3K9me3 peak along with its bound BGCs in Af293 and CEA17. (D) A bar plot showing the expression of BGC genes with (red) or without (blue) H3K9me3 in CEA17. Random remaining non-BGC genes (grey, n = 213) were plotted as control. (E) Bar plots showing the quantified production of SMs pyripyropene A and fumagillin in Af293 and CEA17. The SMs levels were calculated as peak area, and P values were calculated by F test to compare variations. Error bar represent standard derivation and * means P value <0.05; ** means P value <0.01; *** means P value <0.001, **** means P value <0.0001. The pink shade in (A), (B) and (C) is to distinguish data mapping to the CEA17 genome reference.
(TIF)

**S4 Fig. Different TEs distribution between Af293 and CEA17.** (A) A bar plot showing the number of unmapped reads (from Bowtie alignment to the indicated reference genome) matched to TE consensus by Repeatmasker. Error bar represent standard derivation of two biological replicates. (B) Heatmap plots showing the H3 and H3K9me3 deposition at CEA17 transposon loci identified by Repeatmasker and at their boundary regions (+/- 4 kb) and their

expression values (in FPKM). (C) Sector diagrams showing the distribution of H3K9me3 peaks with or without TEs (left panel), and transposons with or without H3K9me3 (right panel) genome wide in CEA17. (D) A bar plot showing the number of DNA and RNA transposons bound by H3K9me3 in CEA17. (E) A bar plot showing the expression level of DNA and RNA transposons in CEA17. (F) A scatter plot showing the relationship between expression level and H3K9me3 modification level at DNA and RNA transposons in CEA17. (G) A bar plot showing the distribution in families of Af293 and CEA17 TEs according to Repeatmasker classification. (H) Heatmap plots showing the LINE transposon level in Af293 and CEA17. (I) A bar plot showing the number of blast hits for LINE transposon in Af293 and CEA17. (J-K) Bar plot showing the genome percentage and mutation reads percentage of (J) Aft1 transposons and (K) 18S rRNAs in Af293 and CEA17. Error bar represent standard derivation of two biological replicates.
(TIF)

**S5 Fig. Different genome stability inferred from comparable published RNAseq data of Af293 and CEA17.** Heatmaps showing the mapped reads in genes of all chromosomes of Af293 clade data (n = 83) and 11 contigs of CEA17 clade data sets (n = 114) from 19 and 28 studies, respectively. Green arrows indicate the genome regions displayed in Fig 6A.
(TIF)

**S6 Fig. The different genome arrangement of evolved Af293GG isolate compared to Af293ATCC isolate.** (A) A heatmap plot showing RNA-seq mapped read density to Chr VIII of Af293 genome reference for the CEA17, Af293GG and Af293ATCC strains. (B) A scheme diagram plot showing the Gene Ontology result of Af293GG up and down-expressed genes measured by RNAP II ChIPseq. (C) Photos showing the 72 hours culture and culture media of CEA17, Af293GG and Af293ATCC. (D) A genome browser screenshot showing the SNP mutation of R202L in Af293GG and Af293ATCC isolates compared to CEA17. (E) A bar plot showing qPCR analysis on the selected genomic loci at the Chr VIII left arm in CEA17, Af293GG and Af293ATCC. Data was plotted as normalized values to Af293ATCC. The qPCRs were processed in two independent replicates and P values were calculated by unpaired t test. Error bar represent standard derivation and * means P value <0.05; ** means P value <0.01; *** means P value <0.005.
(TIF)

**S7 Fig. The induced fumitremorgin production by evolved Af293GG isolate compared to Af293ATCC isolate.** (A) A scheme diagram showing the (top panel) genome distribution of fumitremorgin cluster (BGC 29) genes and (bottom panel) pathway of fumitremorgin A-C biosynthesis. (B) The LC/MS profile of compounds 1–6 in the supernatants of in Af293GG, Af293 and CEA17 strains as shown in (A) and Fig 7J.
(TIF)

**S8 Fig. The evolved Af293GG isolate produced more spores and had higher virulence.** (A) Photos showing the different colony morphologies of CEA17, Af293ATCC and Af293GG isolates. (B) A histogram plot showing the median survival days of the larvae as shown in Fig 7M. Error bar represent standard derivation of three replicates and * means P value <0.05; ** means P value <0.01.
(TIF)

**S9 Fig. Correlation between ChIP-seq and RNA-seq sample replicates.** (A-B) A heatmap (A) and bar plot (B) showing the correlation of ChIP-seq biological replicates. (C) A scatter

plot showing the correlation of RNA-seq biological replicates.
(TIF)

**S1 Table. Genome size of different *A. fumigatus* isolates.**
(XLSX)

**S2 Table. The mapping records of Af293 and CEA17 ChIP-seq data.**
(XLSX)

**S3 Table. The annotated BGC genes information in Af293 and CEA17.**
(XLSX)

**S4 Table. The TE elements identified from RepeatMasker program in Af293 and CEA17.**
(XLSX)

**S5 Table. Sequence variations in the LINE TEs of Af293 and CEA17 compared to the LINE TE consensus sequence.**
(XLSX)

**S6 Table. Genes in the right arm of Chr VIII which were lost in Af293GG isolate.**
(XLSX)

**S7 Table. Gene Ontology analysis results from FungiDB for the Af293GG lost Chr VIII genes.**
(XLSX)

**S8 Table. Differential expressed genes in Af293GG and Af293ATCC isolate detected by RNAP II ChIP-seq.**
(XLSX)

**S9 Table. Gene Ontology analysis results from FungiDB and AspGD for the Af293GG induced and repressed genes.**
(XLSX)

**S10 Table. Genes in the left arm of Chr VIII which had their copy number amplified.**
(XLSX)

**S11 Table. Antibodies used in this project.**
(XLSX)

**S12 Table. Oligos used in this project.**
(XLSX)

**S1 Data. Gene Ontology analysis results from FungiDB of genes in clusters 1–4 as shown S1E Fig.**
(XLSX)

**S2 Data. H3K9me3 peaks detected in Af293 and CEA17 genomes.**
(XLSX)

**S3 Data. Published RNA-seq data of Af293 and CEA17 isolates used in this work.**
(XLSX)

**S4 Data. Conserved 18S rRNA sequence, and the consensus sequences of LINE and Aft1 transposons used for analysis in this work.**
(XLSX)

**S1 Movie. A movie showing the *G. mellonella* movement and myelinization after 3 days of infection with Af293ATCC (first petri dish) or Af293GG (second petri dish).**
(MP4)

## Acknowledgments

This work was performed in part at the High-Performance Computing Cluster (HPCC), which is supported by the Information and Communication Technology Office (ICTO) of the University of Macau. We thank Jacky Chan for technical supports on the HPCC. We acknowledge the technical supports and services from the Genomics, Bioinformatics and Single Cell Analysis and the Proteomics, Metabolomics and Drug Development cores. We also thank Dr. Alessia Buscaino for reading the manuscript draft and the comments and suggestions.

## Author Contributions

**Conceptualization:** Gustavo H. Goldman.

**Data curation:** Lakhansing Pardeshi.

**Formal analysis:** Ana Cristina Colabardini, Fang Wang, Zhengqiang Miao, Lakhansing Pardeshi, Clara Valero, Patrícia Alves de Castro, Daniel Yuri Akiyama, Kaeling Tan, Luisa Czamanski Nora, Rafael Silva-Rocha, Marina Marcet-Houben, Toni Gabaldón, Taicia Fill, Koon Ho Wong.

**Funding acquisition:** Koon Ho Wong, Gustavo H. Goldman.

**Investigation:** Ana Cristina Colabardini, Fang Wang, Zhengqiang Miao, Clara Valero, Daniel Yuri Akiyama, Kaeling Tan, Luisa Czamanski Nora, Rafael Silva-Rocha, Marina Marcet-Houben, Toni Gabaldón.

**Methodology:** Ana Cristina Colabardini, Fang Wang, Zhengqiang Miao, Taicia Fill.

**Project administration:** Gustavo H. Goldman.

**Resources:** Ana Cristina Colabardini.

**Software:** Zhengqiang Miao, Lakhansing Pardeshi.

**Supervision:** Fang Wang, Koon Ho Wong, Gustavo H. Goldman.

**Validation:** Ana Cristina Colabardini, Fang Wang, Zhengqiang Miao.

**Visualization:** Ana Cristina Colabardini, Fang Wang.

**Writing – original draft:** Ana Cristina Colabardini, Fang Wang, Koon Ho Wong, Gustavo H. Goldman.

**Writing – review & editing:** Ana Cristina Colabardini, Fang Wang, Zhengqiang Miao, Lakhansing Pardeshi, Clara Valero, Patrícia Alves de Castro, Daniel Yuri Akiyama, Kaeling Tan, Luisa Czamanski Nora, Rafael Silva-Rocha, Marina Marcet-Houben, Toni Gabaldón, Taicia Fill, Koon Ho Wong, Gustavo H. Goldman.

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
