## [Decision Letter · Decision Letter 0]

25 Jul 2021

Dear Dr Goldman,

Thank you very much for submitting your Research Article entitled 'Chromatin profiling reveals genome stability heterogeneity in clinical isolates of the human pathogen Aspergillus fumigatus' to PLOS Genetics.

The manuscript was fully evaluated at the editorial level and by independent peer reviewers. The reviewers appreciated the attention to an important problem, but raised some substantial concerns about the current manuscript. Based on the reviews, we will not be able to accept this version of the manuscript, but we would be willing to review a much-revised version. We cannot, of course, promise publication at that time.

If you decide to revise the manuscript for further consideration at PLOS Genetics, please aim to resubmit within the next 60 days, unless it will take extra time to address the concerns of the reviewers, in which case we would appreciate an expected resubmission date by email to plosgenetics@plos.org.

[LINK]

We are sorry that we cannot be more positive about your manuscript at this stage. Please do not hesitate to contact us if you have any concerns or questions.

Yours sincerely,

Zachary A Lewis

Guest Editor

PLOS Genetics

Wendy Bickmore

Section Editor: Epigenetics

PLOS Genetics

Reviewer's Responses to Questions

**Comments to the Authors:**

Reviewer #1: The manuscript ‘Chromatin profiling reveals genome stability heterogeneity in clinical isolates of the human pathogen Aspergillus fumigatus’ reports on chromatin profiles for activating (H3K4m3) and repressing (K3K9me3) histone modifications in two A. fumigates isolates. The manuscript shows differences in transposable elements (TEs) and in genome instabilities, the latter contribute to fitness and virulence.

Overall, the quality of the manuscript is good and it reports on an exciting suite of novel data that will be interesting for researchers working on Aspergillus, chromatin biology, and genome evolution, among others. However, I have few major concerns related to the experimental approaches and results, and to which extent the data robustly supports the manuscript’s main conclusions.

Major remarks:

The manuscript shows a) differences in H3K9me3 and TE abundance between different strains, b) differential abundance and activity of specific TEs (LINEs), and c) that strains harbor genomic instability. Based on these observations, the authors conclude that ‘Overall, our work emphasizes the importance of epigenetic modifications in A. fumigatus strain heterogeneity as evolutionary drivers of altered gene expression and genome stability’. While these observations are very interesting, e.g., the chromosomal duplication of chr.8, and clearly correlated, the causative link between chromatin and genome instability has not been established. Knock-out experiments (coupled with ChIP-/RNA-seq) aimed to remove H3K9 methylation are necessary to demonstrate that TE activity is suppressed by H3K9m3 and can be re-established in the absence of this chromatin modification. Furthermore, RNA-seq and H3K9m3 ChIP-seq under different conditions would firmly establish whether TEs, as previously demonstrated for instance for Z. tritici (Fouche et al. 2020), and especially LINEs are displaying differential and H3K9me3 dependent activity (this would also be relevant for the expression of telomere-localized genes, see below). Lastly, the link between H3K9me3 to genome stability needs to be experimentally established. Similar to recent work in in Z. tritici (Möller et al. 2019), Aspergilli with altered H3K9me3 needs to be grown for an extended period of time and re-sequenced to show that i) TEs are re-activated (see above) and ii) that genome instability is increased compared with wild-type strains. In line with this concern, the manuscript’s title does not reflect the conclusions that can be safely drawn from the presented data and analyses, and thus should be toned done at least.

The manuscript reports on differences in TE numbers between isolates, and many of these TEs occur at sub-telomeric regions (Fig.2/4). Fedorova et al. 2008 have previously reported on genomic differences at (sub) telomeric regions of A. fumigatus by aligning the CEA17 genome to the Af293 reference. It is thus unclear if the observed differences in TE abundances are due to technical (assembly) or biological factors. Therefore, the authors need to address the differences in genome assembly quality before drawing firm conclusions on differences in TE content. Fig. 4h shows the number of TEs in different genomic regions. It would be clearer if the figure would display TEs in relation to the size of the respective region as sub-telomeric regions are much smaller compared with the overall chromosome and consequently enrichment of TEs in sub-telomeric regions is not clearly visible.

Some controls/data are missing:

The manuscript does not report any biological replicates for the ChIP-seq experiments to assess the robustness of the observed patterns. It is essential to quantify the biological variation in these experiments (correlation between repeats), especially when quantitatively comparing levels of histone modifications such as shown in Fig. 1.

The manuscript reports on potential centromeric regions using sequencing read coverage as evidence (L298). Are these locations identical to the ones reported by Fedorova et al. 2008? To firmly establish centromeric positions, ChIP-seq experiments targeted at CenH3/CENPA are essential.

Some important references are missing:

The manuscript mentions (L109) that post-translational modification in A. fumigatus and their role in virulence and pathogenicity are unknown. However, several articles have started to explore these links in the couple of years, which need to be discussed and addressed (such as Palmer et. al 2008; Li et al. 2019; Lin et al. 2020).

Further comments:

L67: the suggested contradiction between environmental diversity and the capacity of A. fumigatus to infect patients is not clear. Could the authors please clarify this point?

L282: ‘corresponding to the Af293…’; this part of the sentence is redundant to line 279 and should be removed

L285: The authors observe an interesting relationship between gene expression levels and histone modifications, as previously reported in other fungal system. They also report on the location of suppressed genes (group 3) at H3K9me3 regions at telomeres. Telomeric regions have previously been shown to be transcriptionally dynamic during infections (McDonagh et al. 2008), and it would be very interesting to assess if these H3K9me3 marked genes are expressed during infection. Typically, H3K9me3 has been considered to mark obligatory heterochromatin and facultative heterochromatin such as H3K27me3 is absent in Aspergilli. Thus, this observation would be an interesting observation to be added to the paper.

L632: The manuscript discusses potential reasons for the presence of H3K4me3 is not strictly linked to increased gene expression (L631). One additional explanation is related to the authors’ analyses of only a single activating mark, while many more are chromatin marks associated with active genes are known in Aspergilli and other fungi (see Collemare et al. 2019). Thus, an additional explanation could indeed be, as suggested by the authors, that H3K4 is absent but replaced by another activating mark.

L702: ‘tospeculate’ � ‘to speculate’

L836: the Nanopore genome assembly was finalized using Ragout with a publicly available reference genome as reference strain. Fungal isolates can differ considerably in the chromosomal makeup even between closely related isolates and the authors

L858: how were TE families defined? Did the authors predict them de novo or were these based on previously generated consensus sequences?

L866: the sentence reads odd and needs to be rephrased

Fig. 1a: the ChIP-seq signal higher for one of the isolates compared with the other. To better compare between isolates the authors need to normalize the signal

Fig. 2a: it would be clearer if the H3K9me3 occupancy patterns would be displayed in relationship to scaled chromosomes sizes (i.e., first 10%, 20%, 30%, etc.) to account for differences in chromosome size (chr1 vs. chr8).

Reviewer #2: The manuscript submitted by Colabardini and Wang et al., provide a comparison of two isolates of Aspergillus fumigatus, termed Af293 and CEA17, the causal agent of Invasive Pulmonary Aspergillosis (IPA). To explore differences between the strains, the authors conducted ChIP-seq for H3K4me3 and H3K9me3. The authors mainly addressed how domains of H3K9me3 associated with secondary metabolite clusters in the genome and transposable elements. The authors report that TEs were significantly different between the two lines, with a major difference being Af293 having more LINE retrotransposons. The authors provide some comparative analysis that this increase results in more genomic instability for line Af293. The authors report on a derivative strain of Af293, termed Af293GG that contains a ~320kb deletion, altered SM profile, an amplified region and altered morphology.

While the paper is interesting, it suffers from imprecise characterization and breaks in the logical flow.

Major concerns:

Is there a better control whereby the authors can support their claim that using the Af293 genome for CEA17 data results in accurate conclusions? The authors only report a similar mapping rate, but this is a weak argument. The authors can bolster the methodological approach, and thereby the findings, by providing more evidence the results are not because of using a single reference.

The analysis is not precise in places. Phrases from the authors such as ‘’ the distribution of H3K9me3 peaks [..] of the two isolates was somewhat different’. This is too colloquial and lacks from formal/rigorous analysis. The following conclusion ‘Therefore, these results reveal dissimilarities in the pattern of H3K9me3 deposition on Af293 and CEA17 genomes’ is difficult to assess because the analysis itself is not complete.

There does not appear to be substantial evidence that supports the conclusion that Af293 and CEA17 have different epigenetic patterns (section lines 260-324). For instance, it is not clear what is being shown in Figure2 a, and the analysis is not systematic or complete. Figure 2g and h are just pictures and it is difficult to draw a conclusion about how similar or different they are. The conclusion seems to be more concreate than the supporting data.

A similar problem for the SM analysis, it is not complete, and the data shown do not support the conclusions. How does figure 3a show that two clusters have a similar H3K9me3 profile? Where is data showing the other SMs clusters do not have a similar profile? Are the SM clusters in strain CEA17 well assembled and identifiable?

Why does line 343 say ‘we found only two H3K9me3 peaks located within SM BGCs for both’, while line 359 says ‘Among the H3K9me3-associated clusters, five clusters (BGCs 8, 10, 16, 19 and 25) (Figure 3c) were commonly modified by H3K9me3 between the two isolates.’ ? Is this a difference between H3K9me3 being within the cluster versus close by? It is not clear.

It is very difficult to put much faith in a TE comparison between a well assembled and a less well assembled genome. How does the better reference of Af293 affect the results? The detection of 5x more LINE elements in Af293 may be true, and some additional analysis is provided, such as a qPCR for a specific DNA segment. But then the authors make a significant logical leap with the concluding of the paragraph (lines 455-456), ‘these results imply that Af293 genome may have higher LINE-transposon activity and, hence, more unstable.’

The section using public RNA-seq data for the Af293 strain to check for chromosomal aberrations, detected as loss of read-mapping for contiguous regions, is quite interesting. The important control, similar analysis for strain CEA17, is present. What is not clear is how similar or different are the experiments that make up the two sets of data. Can the authors make a table that summarizes somehow the types of experiments that are captures by the two sets of data. That is, was a given RNAseq bioproject looking at growth and development, mutant analysis, in vitro growth, stress conditions, etc. For instance, data from PRJNA421149 appear to have significantly more lost regions than other data sets. Is the because of the strain used in that lab or because of the experimental conditions from that set of experiments? The alternative hypothesis that is not clearly tested is that the type of experiments that make up the two strains datasets are themselves very different and therefore influence the amount of genomic instability observed.

I do not think there is a very strong link between the data and the overall conclusion. (line 759-761) ‘Overall, our work emphasizes the importance of epigenetic modifications in A. fumigatus strain heterogeneity as evolutionary drivers of altered gene expression and genome stability.’ There is not a strong connection between the epigenetic data collected and the reported structural variation observed in the GG strain’s genome. There is not a rigorous description of epigenetic differences and how this relates to the SV differences. The authors seem to go from the ChIP data, to there are different TEs, to there are different genome stability between the lines. But there is no data for causation. This is needed to ‘emphasize the importance of epigenetic modifications …. as evolutionary drivers’

Minor:

The patterns discussed in panel 1c are clear, but the use of the term ‘positively correlate’ (line 213) has a specific statistical meaning, and no correlation is shown. Either a correlation should be computed and reported, or the term changed. This occurs again at other parts of the manuscript.

There are some duplicated sentences and other English mistakes in the MS. It needs another round of proof-reading.

Reviewer #3: In this manuscript, Colabardini, Goldman and colleagues focus on molecular genetics of the two standard A. fumigatus strains, Af293 and CEA17 (derivative of CEA10). First, the authors perform ChIP-seq experiments to characterize genome-wide distributions of H3K4me3 and H3K9me3. Here the authors find that not all actively transcribed genes are associated with H3K4me3, while not all silent secondary-metabolite clusters are associated with H3K9me3. These observations will surely pave the way for more detailed studies in the future. Second, the authors also analyze TE content and discover that TEs are more prevalent in Af293, where they may actually drive substantial genome instability. To identify instances of large deletions the authors make a clever use of publicly available transcriptomic data-sets.

Overall, this manuscript presents a collection of interesting findings and follow-up experiments concerning the molecular genetics of A. fumigatus.

While the authors identify a clear difference in TE content between the two strains, it would be interesting to see further analysis that might explain this difference. For example, RIP was still reported to occur in A. fumigatus (doi 10.1134/S0026893307060039), despite its largely asexual lifestyle. Is it possible that a lager proportion of TEs in CEA17 (as compared to Af293) are inactivated by RIP?

Minor issues:

Line 563: extra ‘.’ after ‘above’

Line 685: space missing in ‘heresuggest’

Line 702: space missing in ‘tospeculate’

**Have all data underlying the figures and results presented in the manuscript been provided?**

Reviewer #1: Yes

Reviewer #2: **No: **From what I could tell, there is not an SRA entry for PRJNA728823, so I cannot tell if the data are available.

Reviewer #3: Yes

PLOS authors have the option to publish the peer review history of their article (what does this mean?). If published, this will include your full peer review and any attached files.

Reviewer #1: No

Reviewer #2: No

Reviewer #3: No

---

## [Decision Letter · Decision Letter 1]

16 Nov 2021

Dear Dr Goldman,

Thank you very much for submitting your Research Article entitled 'Chromatin profiling reveals heterogeneity in clinical isolates of the human

pathogen Aspergillus fumigatus' to PLOS Genetics.

The manuscript was fully evaluated at the editorial level and by independent peer reviewers. The reviewers appreciated the attention to an important topic but identified some concerns that we ask you address in a revised manuscript. All three reviewers agreed that the study addresses an important topic and that the major conclusions are supported by the data. Reviewer 1 raised a some additional minor concerns. 

We therefore ask you to modify the manuscript according to the review recommendations. Your revisions should address the specific points made by each reviewer.

[LINK]

Yours sincerely,

Zachary A Lewis

Guest Editor

PLOS Genetics

Wendy Bickmore

Section Editor: Epigenetics

PLOS Genetics

Reviewer's Responses to Questions

**Comments to the Authors:**

Reviewer #1: The revised manuscript 'Chromatin profiling reveals heterogeneity in clinical isolates of the human pathogen Aspergillus fumigates' successfully addresses most of my raised comments. As previously mentioned, the manuscript reports on an exciting suite of novel data that will be interesting for researchers working on Aspergillus, chromatin biology, and genome evolution.

I still have few minor remarks to be considered by the authors prior to publication:

The abstract does not explicitly link the differences in chromatin profiles to differences in TE number/activity and ultimately genetic heterogeneity. This link is clearer laid out in the 'Authors summary' (L78) and I would encourage the authors to adjust the Abstract accordingly

L192: Could the authors briefly explain/mention the differences between CEA10 and CEA17?

Fig. 3 shows that some H3K9me3 peaks occur outside and inside of BGC. Could the authors please add the TE track to these pictures to show that these peaks are (not) associated with transposons and other repeats that could explain the different patterns, i.e., association to TEs rather than association to BGCs

The manuscript reports that isolated TEs do not correlate with H3K9me3. However, only peaks called by MACS2 were used rather than assessing the quantitive H3K9me3 levels above the background. This could still reveal an association of this chromatin mark to TEs outside of TE cluster. Thus, I would encourage to briefly address and test this possibility.

L691: the authors discuss the occurrence of RIP and its link to the sexual cycle. Recent work in Z. tritici (Moeller et al. 2021) discusses a process that similar to RIP leads to C->T mutations but operates during mitosis, which might be relevant to mention and discuss here as well.

L703: FtmD should be italic

Reviewer #2: The authors had a number of critical comments from the first round of review, and the re-submission has addressed a number of them. The authors has put a lot of effort into the revisions and improved the manuscript. The overall language has been softened around the causal link between the epigenome and genome stability, but there still remains a meaningful discussion on the topic. There still maybe some errors around the TE analysis due to the differences in assemblies, but this is difficult to avoid and the overall conclusions are likely valid in this reviewers opinion.

Reviewer #3: This reviewer fully supports publication of the revised manuscript in PLoS Genetics.

**Have all data underlying the figures and results presented in the manuscript been provided?**

Reviewer #1: Yes

Reviewer #2: Yes

Reviewer #3: Yes

PLOS authors have the option to publish the peer review history of their article (what does this mean?). If published, this will include your full peer review and any attached files.

Reviewer #1: **Yes: **Michael F Seidl

Reviewer #2: No

Reviewer #3: No

---

## [Editor Report · Decision Letter 2]

17 Dec 2021

Dear Dr Goldman,

We are pleased to inform you that your manuscript entitled "Chromatin profiling reveals heterogeneity in clinical isolates of the human pathogen Aspergillus fumigatus" has been editorially accepted for publication in PLOS Genetics. Congratulations!

Before your submission can be formally accepted and sent to production you will need to complete our formatting changes, which you will receive in a follow up email. Please be aware that it may take several days for you to receive this email; during this time no action is required by you. Please note: the accept date on your published article will reflect the date of this provisional acceptance, but your manuscript will not be scheduled for publication until the required changes have been made.'

Please make the following minor grammatical changes to your manuscript prior to production:

line 152: "The A. fumigatus genome contains an abundance...."

line 172: "TEs" should be "TE"

line 526 "folds" should be "fold"

Yours sincerely,

Zachary A Lewis

Guest Editor

PLOS Genetics

Wendy Bickmore

Section Editor: Epigenetics

PLOS Genetics

Comments from the reviewers (if applicable):

**Data Deposition**

http://datadryad.org/submit?journalID=pgenetics&manu=PGENETICS-D-21-00770R2

**Press Queries**

---

## [Editor Report · Acceptance letter]

7 Jan 2022

PGENETICS-D-21-00770R2 

Chromatin profiling reveals heterogeneity in clinical isolates of the human
pathogen Aspergillus fumigatus 

Dear Dr Goldman, 

We are pleased to inform you that your manuscript entitled "Chromatin profiling reveals heterogeneity in clinical isolates of the human
pathogen Aspergillus fumigatus" has been formally accepted for publication in PLOS Genetics! Your manuscript is now with our production department and you will be notified of the publication date in due course.

With kind regards,

Livia Horvath

PLOS Genetics

On behalf of:
